# Regional Grid Refinement in an Earth System Model: Impacts on the Simulated Greenland Surface Mass Balance

Leonardus van Kampenhout[1], Alan M. Rhoades[2], Adam R. Herrington[3], Colin M. Zarzycki[4], Jan T.M. Lenaerts[5], William J. Sacks[4], and Michiel R. van den Broeke[1]

[1]Institute for Marine and Atmospheric Research Utrecht, Utrecht University, The Netherlands
[2]Lawrence Berkeley National Laboratory, Berkeley CA, USA
[3]Stony Brook University, Stony Brook NY, USA
[4]National Center for Atmospheric Research, Boulder CO, USA
[5]Department of Atmospheric and Oceanic Sciences, University of Colorado, Boulder CO, USA

**Correspondence:** Leo van Kampenhout (L.vankampenhout@uu.nl)

**Abstract.** In this study, the resolution dependence of the simulated Greenland ice sheet (GrIS) surface mass balance (SMB) in the variable-resolution Community Earth System Model (VR-CESM) is investigated. Coupled atmosphere-land simulations are performed on two regionally refined grids over Greenland at 0.5°(∼55 km) and 0.25° (∼28 km), maintaining a quasi-uniform resolution of 1°(∼111 km) over the rest of the globe. On the refined grids, the SMB in the accumulation zone is significantly improved compared to airborne radar and in-situ observations, with a general wetting (more snowfall) at the margins and a drying (less snowfall) in the interior GrIS. Total GrIS precipitation decreases with resolution, which is in line with best-available regional climate model results. In the ablation zone, CESM starts developing a positive SMB bias with increased resolution in some basins, notably in the east and the north. The mismatch in ablation is linked to changes in cloud cover in VR-CESM, and a reduced effectiveness of the elevation classes subgrid parametrization in CESM. Overall, our pilot study demonstrates that variable resolution is a viable new tool in the cryospheric sciences, and could e.g. be used to dynamically downscale SMB in scenarios simulations, or to force dynamical ice sheet models through the CESM coupling framework.

*Copyright statement.* TEXT

## 1 Introduction

The contribution of the Greenland ice sheet (GrIS) to global sea level rise is increasingly determined through its surface mass balance (SMB) (van den Broeke et al., 2016). Accurate estimates of future GrIS SMB are therefore key in providing projections for sea level rise. Arguably the most realistic SMB projections to date are derived from general circulation model (GCM) scenario output downscaled using regional climate models (RCMs — e.g., Rae et al. (2012); van Angelen et al. (2013); Fettweis et al. (2013a); Mottram et al. (2017); Noël et al. (2018)). Compared to GCMs, the regional models offer more sophisticated snow models that have improved representation of albedo, melt, firn densification and refreezing, features that are lacking in

most current GCMs (Ziemen et al., 2014; Helsen et al., 2017). In addition, RCMs typically run at a horizontal grid resolution of $\mathcal{O}(10\ km)$ whereas atmospheric GCMs are typically run using $1°$ or $\mathcal{O}(100\ km)$ grids. RCMs therefore tend to better resolve topographic gradients, which leads to more accurate spatio-temporal distributions in precipitation, wind, cloud cover, and temperature, enabling a detailed comparison to in-situ meteorological data. A fine spatial resolution seems indispensable for resolving narrow ablation zones found around the GrIS margins (Lefebre et al., 2005; Pollard, 2010).

Recently, significant efforts have been invested into making GCMs more suitable for snow and SMB modelling (e.g., Punge et al., 2012; Cullather et al., 2014; Fischer et al., 2014; Helsen et al., 2017; van Kampenhout et al., 2017; Shannon et al., 2019; Alexander et al., 2019). In particular, improvements made to the Community Earth System Model (CESM) include a multilayer snow model with a two-way radiative transfer model for albedo (Flanner and Zender, 2005), enhanced snow density parameterizations (van Kampenhout et al., 2017), and the introduction of multiple elevation classes for downscaling SMB with height (Lipscomb et al., 2013). Still, significant biases remain with respect to RCMs, as the spatial resolution is limited (Vizcaíno et al., 2013; Helsen et al., 2017). Although high-resolution GCM simulations exist (50 km, Delworth et al. (2011); 25 km, Wehner et al. (2014); Small et al. (2014); Bacmeister et al. (2014); 80 km, Müller et al. (2018)) a majority of ongoing modelling experiments, notably the forthcoming CMIP6 experiments (Eyring et al., 2016), maintain a $\sim 1°$ atmosphere grid due to limitations in computational resources.

A middle road may be found in new techniques that apply regional grid refinement within a global climate model. In this approach, a static global mesh is constructed which has increased resolution over a specified region of interest. Over the past five years, progress has been made in developing regional grid refinement in CESM — variable resolution or VR-CESM. To date, studies looked at the effect of grid refinement on the global circulation and climatology (Zarzycki et al., 2015; Gettelman et al., 2018), the effect on tropical cyclones (Zarzycki and Jablonowski, 2014), regional climate in the presence of mountains (Rhoades et al., 2015; Huang et al., 2016; Rhoades et al., 2017), and the scale dependence of the underlying physics (Gettelman et al., 2018; Herrington and Reed, 2018). Compared to RCM downscaling, Huang et al. (2016) notes several advantages of the variable resolution (VR) approach. First, using a unified modelling framework avoids the inconsistencies between RCM and GCM, in particular the different dynamical core and physics that are used. Second, VR allows for two-way interactions (i.e., downstream / upstream effects) between the refinement region and the global domain, which an RCM downscaling approach does not. Finally, some more practical advantages are the attractiveness of operating a single modelling framework, and the relatively low computational cost associated with VR-CESM.

In this paper, we apply regional grid refinement over the Greenland area using VR-CESM, and explore the impacts that the refinement has on GrIS SMB. Two VR meshes are constructed with refined patches centered around the GrIS with 55 km and 28 km resolution, respectively. A 20-year atmosphere-only simulation spanning the historical period (1980-1999) is carried out over each of those grids and is then compared to a reference simulation without refinement, reanalyses data, airborne snow accumulation radar, in-situ SMB measurements, as well as gridded climate data from an RCM. The version of CESM used resembles the recently released CESM version 2 (CESM2), of which a more in-depth evaluation will be published in the near future. Our modelling setup and benchmark data are described in further detail in Section 2. The main findings of this study are presented and discussed in Section 3, and the main conclusions are found in Section 4. Our study is part of an ongoing effort

to improve the representation of ice sheets in CESM (Lipscomb et al., 2013; Vizcaíno et al., 2013; Lenaerts et al., 2016; van Kampenhout et al., 2017).

## 2 Methodology

### 2.1 Modelling setup

The Community Earth System Model (CESM) is a global climate modelling framework comprised of several components, i.e. atmosphere, ocean, land surface, sea ice, and land ice, that may operate partially or fully coupled. When partially coupled, the missing components can be substituted by external data or even inactive (stub) components. Here, we follow the protocol of the Atmospheric Model Intercomparison Project (AMIP, Gates et al. (1999)) and dynamically couple the atmosphere-land components and prescribe ocean and sea ice data at monthly intervals (Hurrell et al., 2008). Our three AMIP-style CESM

simulations are carried out over the years 1980-1999, a period prior to the onset of persistent circulation change and a strong decline in GrIS SMB in the 2000s (Fettweis et al., 2013b; van den Broeke et al., 2016). Aerosol and trace gas emissions are taken as observed.

The atmosphere component used is the Community Atmosphere Model version 5.4 (CAM5.4, Neale et al., 2012) with the spectral element dynamical core (CAM-SE, Dennis et al., 2012; Lauritzen et al., 2018), the only dynamical core currently in

CESM supporting VR capabilities (Zarzycki et al., 2014). VR capabilities in CAM6, the new atmosphere model in CESM version 2 (CESM2, www.cesm.ucar.edu/models/cesm2.0) were still under beta testing at the start of our study, which explains the slightly older model version of CAM. Our model configuration broadly follows that of Zarzycki and Jablonowski (2014), with a few modifications. These include updated rain and snowfall microphysics (MG2, Gettelman and Morrison (2014)), a new dry-mass, floating Lagrangian, vertical coordinate with 32 levels in the vertical, and slightly reduced horizontal diffusion

in the SE dynamical core (Lauritzen et al., 2018). Further, we adopt the Beljaars et al. (2004) orographic drag parameterization which replaces the turbulent mountain stress (TMS) scheme of CESM1 (Neale et al., 2012) in order to achieve more realistic (higher) wind speeds over the ice sheets. Physics tuning coefficients were set to default values specified in the supported CAM5 release.

### 2.2 Grids

Our reference simulation, referred to as Uniform CESM, uses a standard cubed sphere grid which is quasi-uniform at $1°$ ($\sim$111 km) resolution globally (Evans et al., 2012). The first non-uniform VR mesh has a refined patch of $0.5°$ ($\sim$55 km) over the greater Greenland region and is referred to as VR-CESM55. The patch was constructed such that the boundary of the patch always extends at least six spectral elements away from the Greenland coast (Figure 1). This buffer region is intended to allow incoming "low-resolution" storms to develop finer-scale structures after entering the VR zone and prior to making landfall

(Matte et al., 2017). The second VR mesh is constructed off the VR-CESM55 grid, yet features a second level of refinement at $0.25°$ ($\sim$28 km) inside the first level. This second patch was chosen such that, again, the boundary extends at least six spectral

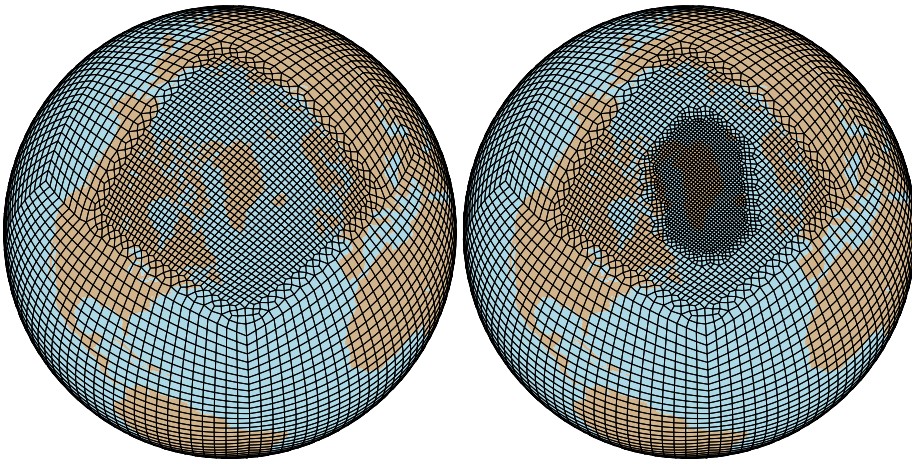

**Figure 1.** Computational domains of experiments VR-CESM55 (left) and VR-CESM28 (right). Each spectral element visible here contains an additional 3-by-3 grid of points, the exact position of which are determined by the spectral element method (Zarzycki and Jablonowski, 2014).

elements away from the Greenland coast. The simulation on this grid is referred to as VR-CESM28 and both VR grids were constructed using SQuadGen (Ullrich, 2014).

Topographic height over Greenland was interpolated from the 4 km CISM ice sheet domain, which in turn has been derived from the 90 m Greenland Ice Mapping Project product (GIMP, Howat et al. (2014)). Topography is static in time – ice sheet dynamics are not active in this configuration – a reasonable assumption for the decadal length simulations presented in this paper. The new ice topography was spliced into the global topography, similar to what is done in two-way coupled setups where ice sheet dynamics are turned on[1]. Due to the hybrid sigma vertical coordinate system implemented in CAM-SE, a differential smoothing procedure was applied to ensure numerical stability and realistic flow, as described by Zarzycki et al. (2015). Subgrid height variances, used by the orographic drag parameterization, are consistently recomputed as a residual of the smoothed topography. At increased resolutions, less smoothing is applied in total leading to a more detailed and accurate representation of topography, see Figure 2. Mean topographic height over the GrIS – as seen by CAM – is 1884 m (Uniform CESM), 2009 m (VR-CESM55) and 2058 m (VR-CESM28), respectively. The feature most prominently improving is the southern ice dome, that "rises up" from ∼2300 m at 111 km to ∼2900 m at 28 km (Figure 2). Furthermore, the 28 km resolution seems sufficiently detailed to start resolving some of the fjord structures, especially in the east. The non-zero topographic heights over open ocean in Figure 2 are explained by the differential smoothing procedure.

The CAM physics (dynamics) time steps for Uniform CESM was 1800 (150) seconds. For the VR-CESM runs, the physics time step was set to 450 s and the CAM dynamics time steps were scaled with horizontal resolution with VR-CESM55 at 150 s and VR-CESM28 at 75 s. Hyperviscosity coefficients are scaled by the grid-resolution (element dimensions) for

---

[1]For details, see the CESM Land Ice Documentation and User Guide, https://escomp.github.io/cism-docs/cism-in-cesm/versions/release-cesm2.0/html/clm-cism-coupling.html

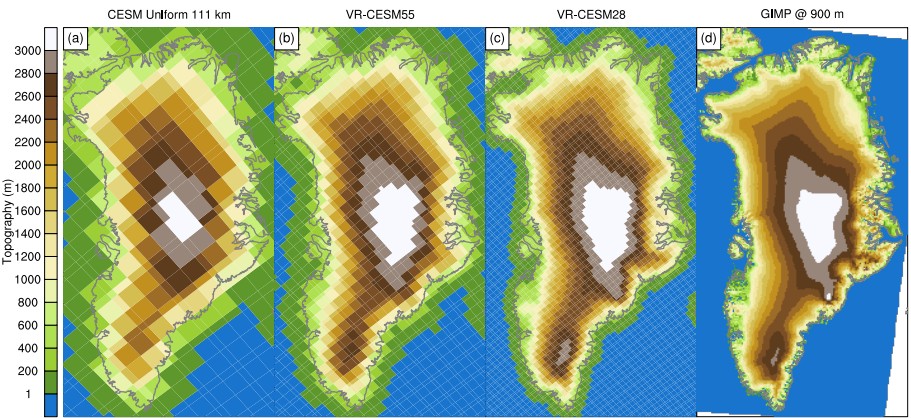

**Figure 2.** Topographic height in the three CESM simulations. For plotting purposes, spectral element node heights are displayed on control volumes equal to the area that they represent. The control volumes are identical to those used by the CESM coupler to conserve mass and energy. For reference, topographic height according to the Greenland Ice Mapping Project (GIMP, Howat et al. (2014)) is shown, which has been upscaled to 900 m.

numerical stability and filter undesirable numerical artifacts (Guba et al., 2014). Here the scaling is such that the hyperviscosity coefficients are reduced by an order of magnitude for each doubling of the resolution (Zarzycki and Jablonowski, 2014). Some minor grid imprinting was noted in the grid transition zone over distorted SE elements. It is deemed unlikely, however, that these small, local anomalies materially impact the large-scale synoptic flow in the interior of the domain.

## 2.3 Land surface model

CAM is coupled to the Community Land Model (CLM) version 5.0, which incorporates several important bug fixes and snow parameters updates for CESM2. CLM simulates the interaction of the atmosphere with the land surface, notably the surface energy balance and hydrological processes such as interception by canopy, throughfall, infiltration, and runoff (Oleson, 2013). For radiation calculations over snow, the two-way radiative transfer model SNICAR is used (Flanner and Zender, 2005). The snow pack hydrological and thermal evolution is modelled as a one-dimensional column, which can reach depths of up to 10 metres of w.e. (water-equivalent), with up to 12 layers. Several snow model modifications have been implemented specifically for ice sheets, such as wind-dependent fresh snow density and wind driven snow compaction (van Kampenhout et al., 2017) and temperature dependent fresh snow grain size. Bare ice albedo is assumed constant, and is set to 0.50 (0.30) for the visible (near-infrared) spectrum, reflecting the distinction made between these two shortwave bands in CLM (Oleson, 2013). Due to a CAM model bias leading to excessive rainfall over the GrIS, the phase of precipitation is recomputed in CLM using a simple temperature threshold. Over non-glacier (glacier) landunits, a threshold temperature of 0 °C (-2 °C) is used for solid precipitation and +2 °C (0 °C) for liquid precipitation, with a linear ramp in between, which has proven effective at removing most of the bias.

Over glaciated grid cells, CLM maintains 10 different elevation classes (ECs) in order to more accurately capture SMB gradients in the ablation zones (Lipscomb et al., 2013). ECs are implemented as independent CLM columns, i.e. each one maintains its unique snow pack, temperature, hydrology, and snow grain size. The weight assigned to each EC is proportional to the subgrid topography present in the CLM grid cell and classes with zero weights are considered "virtual" and do not contribute to the grid cell average, i.e. the value that the atmosphere receives back. ECs are associated with elevation bins, but their specific elevation is determined by the distribution of topography within that bin (see (Lipscomb et al., 2013) for more details). Atmospheric surface temperature and downwelling longwave radiation are downscaled from the mean CLM grid cell elevation to the EC elevation, using global lapse rates. The lapse rate for temperature is 6 K km$^{-1}$ (Lipscomb et al. (2013)) and that for longwave radiation 32 W m$^{-2}$ km$^{-1}$ (Figure 6 in Van Tricht et al. (2016a)), with a grid cell normalization to keep the mean value conserved. Further, specific humidity is downscaled following the assumption that relative humidity remains constant with height. Low lying ECs in the ablation zone typically experience larger melt rates because of enhanced sensible heat and downwelling longwave fluxes, and are therefore instrumental to resolving the narrow ablation zones of Greenland (Figure S1, Supplementary Material).

The Community Ice Sheet Model (CISM, Lipscomb et al. (2019)) is active as a diagnostic component in our simulations in order to benefit from the EC infrastructure. CISM operates on a 4 km Cartesian grid, and receives its surface forcing (temperature, SMB) from CLM through a bi-linear downscaling procedure in order to prevent artificial jumps between grid cells (Leguy et al., 2018). In this study, we deviate from the standard CLM definition of SMB, which does not take into account changes in snow pack height, in favour of the definition that is common to glaciology, in absence of redistribution/erosion by drifting snow

$$SMB = Precipitation - Sublimation - Runoff. \tag{1}$$

For the remainder of the paper, modelled accumulation (ablation) is defined as modelled SMB for locations where SMB > 0 (SMB < 0).

Following Rhoades et al. (2018), the distribution of plant functional types in CLM is assumed constant at year 2000 values for all simulations. As the main focus of this work is on precipitation and snow cover in non-vegetated regions, we argue this assumption has a negligible impact on our results.

## 2.4 Initialisation

In glaciated regions, the subsurface conductive heat flux at the ice sheet surface is potentially large due to the high thermal conductivity of ice. To avoid unrealistic energy losses or gains from the subsurface, one should start with ice that is in thermal equilibrium with the ambient climate. In our modelling setup, however, a sufficiently long spinup period to achieve such equilibrium was not feasible due to computational constraints. Instead, it was decided to initialise deep ice temperature from values close to observed, in this case 10 m firn temperatures from a firn densification model, forced by RCM-downscaled reanalysis data (Ligtenberg et al., 2018). A nearest neighbour procedure was followed to interpolate ice temperature from the 11 km firn model to the different resolutions used in this study.

Below 1774 m in elevation (which corresponds to the highest GrIS elevation where SMB=0 in the RACMO2 climatology), the initial snow amount was set to a maximum value of 100 mm w.e. to avoid snow cover hysteresis resulting from errors in the interpolated initial conditions. A spinup simulation was then carried out to rebuild snow packs in CLM columns below this reset altitude, at least where CESM climate allows it. The relevance of this spinup is two-fold: (1) the dependence of fractional snow cover on snow height (Swenson and Lawrence, 2012), (2) the refreezing capacity of the snow pack. For both of these, a period of 5 years was deemed sufficiently long to capture the first-order effect. Nonetheless, it is recognised that the resulting snow depth distribution over the GrIS contains an artificial jump at 1774 m.

## 2.5 Performance

All simulations have been performed on NCAR's supercomputing facility "Cheyenne" in Wyoming, USA, which is equipped with Intel Broadwell processors. No real load-balancing was needed since the active components (i.e., CAM, CLM, CISM, and coupler) perform well when sharing all the available cores. On 1800 cores (or 50 compute nodes) the cost of Uniform CESM at 1° (48,602 CAM-SE grid points) amounts to ~1070 core hours per simulated year. Keeping the number of cores the same, this cost was tripled to 3250 core hours for the VR-CESM55 simulation with the refined patch of 0.5° (59,402 CAM-SE grid points), and quadrupled to ~4300 core hours for the VR-CESM28 simulation with the additional 0.25° patch (69,887 CAM-SE grid points). By comparison, the computational cost of limited area model RACMO2 at 11 km is ~6800 core hours per simulated year (Brice Noël, pers. comm.). The throughput was ~25, ~13 and ~10 simulated years per day for Uniform CESM, VR-CESM55 and VR-CESM28, respectively.

## 2.6 Reference data

Output from the three CESM simulations is interpreted using reference data from a variety of sources. The evaluation of the climate at synoptic scales is supported by atmospheric reanalyses, i.e. hindcast climate models that employ data assimilation to match the observed state of atmosphere as close as possible. In particular, temperature and geopotential height from the European Centre for Medium-Range Weather Forecasts Reanalysis (ERA-Interim, Dee et al. (2011)) and the Modern-Era Retrospective Analysis for Research and Applications-2 (MERRA2, Molod et al. (2015)) products are used.

For evaluation of GrIS near-surface climate and surface mass balance, data from the Royal Netherlands Meteorological Institute (KNMI) regional atmospheric climate model (RACMO) version 2.3p2 (RACMO2 hereafter) are used. RACMO2 is a state-of-the-art polar climate model that has been extensively evaluated over the GrIS (Noël et al., 2018, 2015) and compares favourably to observations. At its lateral boundaries, RACMO2 was forced using ERA-Interim data and the native spatial resolution of the data is 11 km. When appropriate, however, the statistically downscaled product at 1 km is used, which better resolves narrow ablation zones and low-lying regions (Noël et al., 2016, 2018). We argue that it is fair to compare VR-CESM directly to the downscaled 1 km RACMO2 product as (i) CESM also performs on-line downscaling using the semi-statistical elevation classes (Section 2.3), and (ii) best-estimate data is preferred in order to identify either model improvements or regressions, in line with the purpose of this paper. Still, these best-estimate benchmark data are subject to some uncertainty. Noël et al. (2018) characterise the native spatial resolution of 11 km as a source of model uncertainty, as well as the representation

of surface roughness and surface albedo. Two prime uncertainties in the RACMO2 downscaling procedure arise from the bare ice albedo used to correct runoff, and the ice sheet extent (Noël et al., 2016).

Field data analysis has been carried out through the Land Ice Verification & Validation Toolkit (LIVVkit), an open source software package designed for evaluating ice sheet models and their forcing (Kennedy et al., 2017; Evans et al., 2018). Three observational SMB datasets available for the GrIS are used: (i) airborne radar, (ii) field accumulation (SMB > 0) measurements, and (iii) field ablation (SMB < 0) measurements (Evans et al., 2018). The airborne radar data stems from NASA's Operation IceBridge and covers most of the GrIS interior. The raw data, as described by Lewis et al. (2016), provides seasonal accumulation estimates for a given pixel, uniquely determined by its latitude and longitude. Following Evans et al. (2018), a simple time average is applied over all available periods for each record to yield a single accumulation value (in mm w.e. yr$^{-1}$) per location. The resulting number of IceBridge data points is a sizeable 18,968, which means that the spatial density is quite high over the radar transects. During the evaluation, a nearest neighbour method is used to determine the model cell closest to each observation.

The in-situ field accumulation dataset is a compilation of different field campaigns carried out in the GrIS accumulation zone (Cogley, 2004; Bales et al., 2009; Evans et al., 2018). Only records that have been retrieved using firn cores, snow pits or stake measurements are included in the evaluation. Moreover, if there are multiple measurements at one location then the data is averaged in time to yield a climatological SMB estimate for that location. In total, the number of accumulation zone measurements is 421. The in-situ field ablation dataset is a subset of the compilation of GrIS ablation zone SMB measurements by Machguth et al. (2016). Again, each record location is averaged in time to yield an annual SMB estimate. Only records that are on the CISM ice mask and have a record length equal or close (i.e., within a 5% difference) to a full year are kept, which brings the total number of records down from 627 to 163, spread over 22 rather than 46 glaciers. It is important to mention at this point that the spatial coverage of the ablation zone measurements is quite sparse. Indeed, Figure 1 in Evans et al. (2018) illustrates that all in-situ ablation data stem from merely 8 transects in total.

## 3 Results and Discussion

### 3.1 Large-scale circulation

We start with a comparison of modelled mid-troposphere climate to reanalyses data, which serves two purposes. First, it is useful to identify any significant climatic biases that CESM possesses, which could aid in interpreting e.g. snow melt rates later on. Second, the VR approach allows for feedbacks between the domain of interest and the global climate system, in contrast to dynamical downscaling using RCMs. One such feedback could be changes to the strength and location of planetary waves both in and outside the VR domain, due to the higher and steeper topography (Figures 1 and 2). If such upstream / downstream dynamical effects are present in our modelling setup they would make an imprint on mid-tropospheric climate on a hemispheric scale.

CESM geopotential height (Z500) at 500 hPa is compared against ERA-Interim over the period 1980-1999. Note that the choice of ERA-Interim versus MERRA-2 does not impact our results much, so only ERA-Interim is shown. In boreal summer,

the season most relevant to GrIS SMB, anomaly maps of Z500 display consistent patterns across all three CESM simulations (Figure 3a-c). A positive height anomaly is found over the Arctic ocean, which is most pronounced in the Uniform CESM 111 km simulation. It is surrounded by a band of negative height anomalies in all three simulations, with one of the minima approximately centred over Iceland / South Greenland, indicating more cyclonic flow over the GrIS in CESM. At mid-latitudes,

positive height anomalies are found instead, indicating more anti-cyclonic flow there. Over the region 55-90N, the Z500 root mean squared error (RMSE) is decreased from 3.6 dam (Uniform CESM) to 3.5 dam (-2%, VR-CESM55) and 3.3 dam (-8%, VR-CESM28), which could signal minor benefits of the grid refinement on resolving the large-scale circulation. However, no VR-CESM grid point within the domain of interest can be significantly differentiated from Uniform CESM (t-test, p < 0.05). Furthermore, similar decreases in RMSE are not consistently found in the other seasons (not shown) so these changes are

attributed to internal variability.

Similarly, anomalies of 500 hPa air temperature (T500) with respect to ERA-Interim are computed (Figure 3d-f). Major features in T500 are again shared by the three simulations, such as a cold bias exceeding 0.75 °C over Russia, which is most pronounced in VR-CESM55 and VR-CESM28. A slight June-July-August (JJA) warm bias of around 0.5 - 1 °C is indicated over the Arctic Ocean and Northern Greenland in all three simulations. Over the region 55-90N, RMSE is decreased from 0.74

°C (Uniform CESM) to 0.68 °C (-8%, VR-CESM55) and 0.63 °C (-14 %, VR-CESM28). Similar improvements are found in March-April-May (MAM, Figure S1) but not in the other two seasons (Figures S2 and S3). Some point-by-point significance is found between VR-CESM and Uniform, albeit not over the Greenland area (Figure 3d-f).

To conclude, heights at 500 hPa seem not substantially affected by the enhanced resolution and topography in VR-CESM. In all three CESM simulations, more cyclonic flow is indicated over Greenland with respect to ERA-Interim. Temperature at

500 hPa demonstrates a weakly positive bias in CESM, and shows no significant change with refinement over the GrIS. A weak signal cannot be excluded, however, as it may remain undetected by the Student's t-test due to the relatively short sample period of 20 years.

## 3.2 Precipitation

Both the steep edges of ice sheets as well as topographic promontories are effective drivers of orographic precipitation, as is

e.g. apparent from the RACMO2 precipitation field (Figure 4d). The largest source of moisture is the North Atlantic basin, which is connected to Greenland by large-scale storm systems (Sodemann et al., 2008). Cyclonic activity associated with the persistent Icelandic Low drives warm and moist air onto land from the south-east, resulting in strong orographic uplift which causes rapid cooling, condensation, and precipitation. By comparison, northern Greenland is much drier with accumulation rates locally below 150 mm yr$^{-1}$ (Cogley, 2004).

Since orographic precipitation is dominant over southern Greenland, it is not surprising that we find significant improvements with increasing resolution (Figure 4), compared with RACMO2. At uniform 111 km resolution, CESM correctly predicts a band of high (> 1000 mm w.e. yr$^{-1}$) precipitation rates in the south-east, however it extends too far into the interior (Figure 4a). This is attributed to the fact that the poorly resolved topography is ∼600 m lower in the model than in reality (Figure 2a) and that topographic gradients are smoothed out, which weakens the effect of orographic uplift. The VR-CESM55 result (Figure

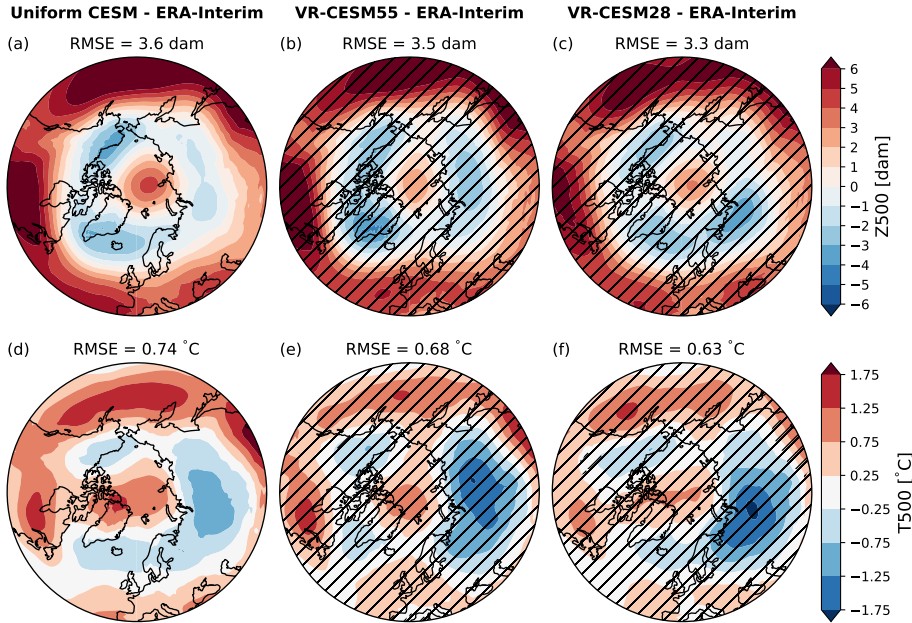

**Figure 3.** Mean summer (JJA) anomalies of 500 hPa geopotential height (Z500, panels a-c) and air temperature (T500, panels d-f) with respect to ERA-Interim over the period 1980-1999. Shown is 55-90N, the same region over which the area-weighted RMSE was calculated that is listed above each panel. Hatching in panel f indicates that the VR-CESM simulation is significantly different ($p < 0.05$) from Uniform CESM. No significance was found in panels b, c, and e. Prior to subtraction, all data have been regridded to a common regular mesh of 1° using bi-linear interpolation.

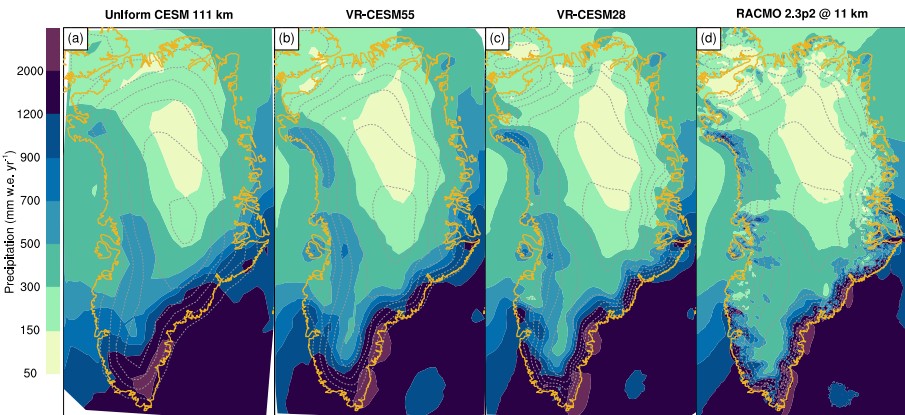

**Figure 4.** Spatial distribution of mean annual precipitation over Greenland. CESM data are displayed at the native CAM resolution for the period 1980-1999. RACMO2 data are shown at native 11 km resolution for the same period. Coastlines and 500 m elevation contours are overlain in orange and grey, respectively. Note the non-linear colour scale.

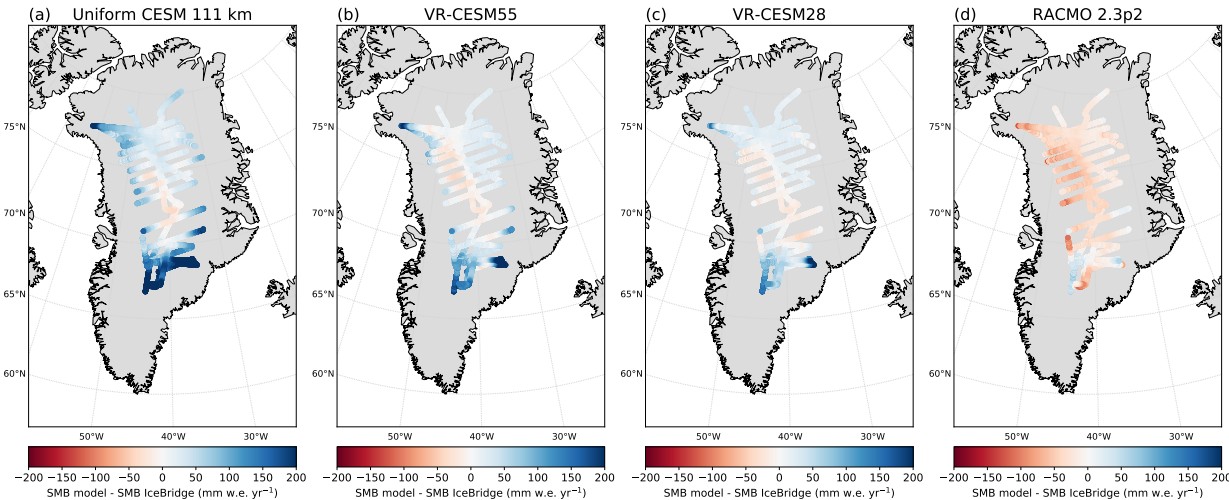

**Figure 5.** SMB differences between IceBridge radar (mean over available period) and model climatology (1980-1999). Blue (red) colours indicate that the model is wetter (dryer) than observations.

4b) shows that this is mostly a resolution issue as the band of high precipitation rates is more confined to the low-lying areas and slopes, similar to RACMO2. Other effects that can be seen in this VR-CESM55 result are the emergence of orographic precipitation in other locations around the margins, albeit weak, and a general drying of the northern interior. In VR-CESM28, similar resolution dependent patterns continue to emerge, with even stronger orographic precipitation and more pronounced drying in the north (Figure 4c). Integrated over the entire GrIS, including peripheral glaciers and ice caps, precipitation is reduced from $946 \pm 107$ Gt yr$^{-1}$ (Uniform CESM) to $870 \pm 72$ (VR-CESM55) and $821 \pm 62$ Gt yr$^{-1}$ (VR-CESM28). By comparison, RACMO2 simulates a mean annual precipitation flux of $743 \pm 64$ Gt yr$^{-1}$ over these glaciated areas. Both the improved patterns (Figure 4) and the more reasonable integrated amount of precipitation over the GrIS are positive results for the application of VR-CESM to this region.

## 3.3 IceBridge

Operation IceBridge accumulation data is used to further quantify the effect of the improved precipitation patterns on SMB. As described in Section 2.3, CESM SMB is downscaled to 4 km using the EC method and averaged over the period 1980-1999, prior to comparison to the processed IceBridge SMB samples (Section 2.6). Figure 5 displays the resulting SMB anomalies in mm w.e. yr$^{-1}$. As can be seen, the comparison with IceBridge radar data supports the pattern of interior drying with increasing resolution. In Uniform CESM at 111 km resolution, we find a mean wet bias of 81 mm w.e. yr$^{-1}$ which is most pronounced in regions near the edges of the IceBridge domain (Figure 5a). The strongest bias is found in the south, where absolute precipitation rates are highest (Figure 4) and any relative error will consequently lead to a larger absolute error. With increasing resolution, the mean bias drops from 81 to 37 mm w.e. yr$^{-1}$ (VR-CESM55) and 24 mm w.e. yr$^{-1}$ (VR-CESM28), which suggests that the largest improvement is made going from 111 km to 55 km (Figure 1). The largest SMB differences remain to

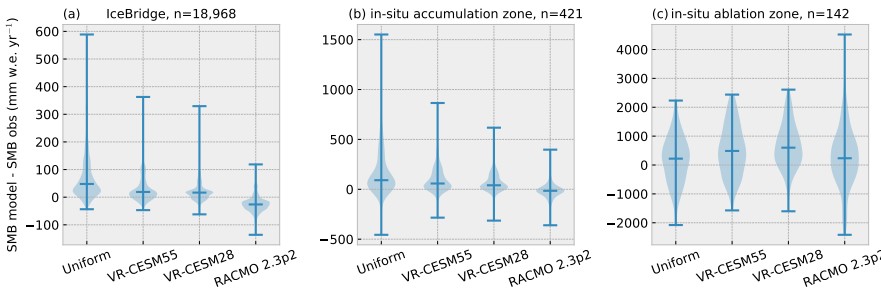

**Figure 6.** Point-by-point SMB differences between model and reference observations. Shading indicates the distribution, and horizontal line segments indicate maximum, median, and minimum value. Model climatologies have been computed over the period 1980-1999.

**Table 1.** Selected statistics of CESM climatological SMB (downscaled to 4 km) and RACMO2 climatological SMB (downscaled to 1 km) with respect to IceBridge radar data, accumulation zone sites, and ablation zone sites. Shown are mean bias, coefficient of determination, and root mean square error. Model climatologies have been computed over the period 1980-1999, which not necessarily overlaps with the date of each measurement.

| | IceBridge ($n = 18,968$) | | | Acc. sites ($n = 421$) | | | |
|---|---|---|---|---|---|---|---|
| | bias (mm w.e. yr$^{-1}$) | $r^2$ | RMSE (mm w.e. yr$^{-1}$) | bias (mm w.e. yr$^{-1}$) | $r^2$ | RMSE (mm w.e. yr$^{-1}$) | bias (m |
| Uniform CESM 1° | 81 | 0.78 | 126 | 187 | 0.61 | 319 | |
| VR-CESM55 | 37 | 0.88 | 68 | 105 | 0.74 | 172 | |
| VR-CESM28 | 24 | 0.92 | 46 | 71 | 0.79 | 124 | |
| RACMO2.3p2 | -25 | 0.94 | 38 | -13 | 0.71 | 91 | |

be found near the margins of the IceBridge domain (Figure 5b-c). The spread in SMB anomaly also decreases with resolution, which can be visually seen as a narrowing of the SMB anomaly distribution in Figure 6a. As a measure for this spread, the difference between the 95th percentile and the 5th percentile falls from 308 mm w.e. yr$^{-1}$ (Uniform CESM) to 178 mm w.e. yr$^{-1}$ (VR-CESM55) and 115 mm w.e. yr$^{-1}$ (VR-CESM28), respectively. As another measure, the RMSE decreases from 126 mm w.e. yr$^{-1}$ (Uniform CESM) to 68 mm w.e. yr$^{-1}$ (-46%, VR-CESM55) and 46 mm w.e. yr$^{-1}$ (-64%, VR-CESM28). At the same time, the spatial correlation is substantially enhanced ($r^2$, Table 1). The bias and RMSE of RACMO2 are -25 mm w.e. yr$^{-1}$ and 38 mm w.e. yr$^{-1}$, respectively, which suggests a dry bias in RACMO2 (Figure 5d). We conclude that based these statistics, VR-CESM28 performs on-par with RACMO2 (Table 1).

### 3.4 Accumulation sites

A similar analysis is carried out for the in-situ accumulation zone observations. Compared to the airborne radar data, these measurements cover a greater portion of the GrIS, including the southern dome (cf. Figure 1 in Evans et al., 2018), which should make it more representative of the GrIS as a whole. As before, the greatest absolute improvement is found in the

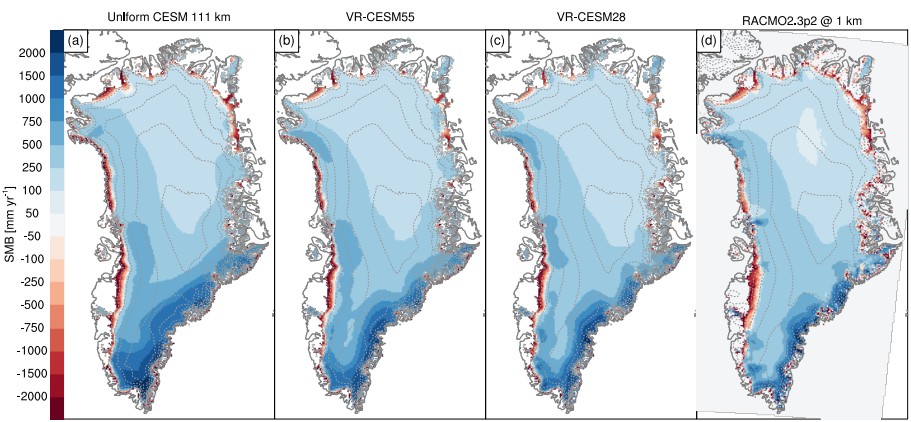

**Figure 7.** Mean annual SMB in mm w.e. yr$^{-1}$. All CESM data are downscaled to 4 km CISM resolution for the period 1980-1999. RACMO2 data have been statistically downscaled from 11 to 1 km. Note the non-linear colour scale.

doubling of resolution from 111 km to 55 km, with smaller benefits going further to 28 km (Figure 6b and Table 1). The mean bias substantially reduces from 187 mm w.e. yr$^{-1}$ (Uniform CESM) to 105 mm w.e. yr$^{-1}$ (-44%, VR-CESM55) and 71 mm w.e. yr$^{-1}$ (-62%, VR-CESM28) and the RMSE reduces from 319 mm w.e. yr$^{-1}$ (Uniform CESM) to 172 mm w.e. yr$^{-1}$ (-46%, VR-CESM55) and 124 (-61%, VR-CESM28). A small positive accumulation bias remains even in the highest

resolution run (VR-CESM28), a bias that is not apparent in the RACMO2 data (Table 1). For RACMO2, the bias and RMSE values are similar to those mentioned by Noël et al. (2018), who report in their Figure 11a an accumulation zone mean bias of -22 mm w.e. yr$^{-1}$ (here: -13 mm w.e. yr$^{-1}$) and an RMSE of 72 mm w.e. yr$^{-1}$ (here: 91 mm w.e. yr$^{-1}$). Our $r^2$ is slightly lower, however, 0.71 against 0.85. These differences can be explained by the different methodology used. Namely, Noël et al. (2018) correlate SMB values based off daily data, thus reflecting the meteorological conditions during which the measurement

was made, whereas here we compare climatological averages of the model to each measurement, which introduces additional noise in the comparison.

### 3.5 Ablation sites

High up on the ice sheet, and thus deep into the accumulation zone, SMB is dominated by snowfall. In the ablation zone, by contrast, there is a delicate balance between different factors — snowfall, sublimation, snowmelt, refreezing, and runoff

— that complicates SMB modelling. Furthermore, SMB gradients are typically much stronger in the ablation zone than they are in the accumulation zone, mainly due to steep topography and non-linearity of SMB with height (Figure 7). Therefore, as one expects, CESM model skill in the ablation zone is lower than in the accumulation zone, signalled by a larger spread and modelling biases exceeding 1000 mm w.e. yr$^{-1}$ at many locations (Figure 6c). Nonetheless, ablation zones are mostly predicted in the right locations (Figure 7), owing to the EC downscaling (Section 2.3) that is active in all simulations.

In contrast to the accumulation zone and somewhat surprisingly, model skill in the ablation zone does not improve with resolution (Table 1). The mean bias grows from 170 mm w.e. yr$^{-1}$ (Uniform CESM) to 462 mm w.e. yr$^{-1}$ (VR-CESM55) and

**Table 2.** Mean GrIS mass fluxes for the period 1980-1999 in gigatonnes per year with standard deviation between brackets. The area of integration is listed in the first column and includes peripheral glaciers and ice caps (GIC). CESM data are integrated at the native resolution with elevation class weighing. The statistically downscaled 1 km RACMO2.3p2 data is averaged over the same period and described in Noël et al. (2018). RACMO2 does not differentiate between snow and ice melt in its output files so only total melt is reported.

| Model name | Ice area | Precipitation | Ice melt | Total melt | Refreezing | Runoff | Sublimation | SMB |
|---|---|---|---|---|---|---|---|---|
| | km$^2$ | Gt yr$^{-1}$ | Gt yr$^{-1}$ | Gt yr$^{-1}$ | Gt yr$^{-1}$ | Gt yr$^{-1}$ | Gt yr$^{-1}$ | Gt yr$^{-1}$ |
| *native ice sheet extent, including GIC* | | | | | | | | |
| Uniform CESM 1° | 1,812,467 | 946 (107) | 217 (48) | 468 (100) | 178 (43) | 349 (67) | 28 (3) | 567 (129) |
| VR-CESM55 | 1,812,254 | 870 (72) | 146 (25) | 387 (70) | 185 (39) | 260 (42) | 39 (3) | 571 (75) |
| VR-CESM28 | 1,812,254 | 821 (62) | 131 (34) | 377 (73) | 195 (35) | 239 (47) | 44 (2) | 538 (87) |
| RACMO2 | 1,761,475 | 743 (64) | - | 577 (81) | 309 (27) | 344 (68) | 33 (2) | 365 (109) |
| *contiguous GrIS extent* | | | | | | | | |
| Uniform CESM 1° | 1,705,508 | 893 (104) | 157 (37) | 361 (85) | 150 (40) | 258 (53) | 26 (3) | 610 (116) |
| VR-CESM55 | 1,692,629 | 796 (69) | 115 (20) | 314 (62) | 159 (37) | 203 (34) | 36 (3) | 557 (71) |
| VR-CESM28 | 1,697,054 | 745 (59) | 105 (28) | 304 (63) | 165 (33) | 184 (38) | 40 (2) | 521 (77) |
| RACMO2 | 1,700,772 | 707 (61) | - | 509 (72) | 263 (25) | 298 (58) | 32 (2) | 376 (99) |

600 mm w.e. yr$^{-1}$ (VR-CESM28), which are substantial increases of +172% and +253%, respectively. The model spread is only marginally detoriated, and RMSE ranges 793 - 951 mm w.e. yr$^{-1}$ for all simulations (Table 1). The ablation statistics of the overall best simulation (Uniform CESM) are comparable to those of RACMO2 which are, analogous to CESM, computed using a 1980-1999 climatology. The bias, $r^2$, and RMSE of RACMO2 are considerably worse than those reported by Noël

et al. (2018), who find a bias of 120 mm w.e. yr$^{-1}$ (here: 160 mm w.e. yr$^{-1}$), an $r^2$ of 0.72 (here: 0.54), and RMSE of 870 mm w.e. yr$^{-1}$ (here: 922 mm w.e. yr$^{-1}$) in their ablation zone comparison with similar data (their Figure 11c). Again, this is explained by the different methodology used. In particular, we believe that some extreme ablation events that happened after the year 2000 are not well captured by the climatological mean of the two 20th century decades considered here. When the period of the RACMO2 climatology is changed to 1995-2017, we find a bias of -9 mm w.e. yr$^{-1}$, an $r^2$ of 0.69, and RMSE of

722 mm w.e. yr$^{-1}$, which confirms that the time frame used is a crucial factor. Overall, we conclude that both VR-CESM55 and VR-CESM28, despite their higher resolution over the GrIS, fail to recover in-situ ablation rates with a skill similar or better than the reference simulation. Instead, a strong positive SMB bias develops in some ablation zone sites, suggesting too little runoff and/or too much precipitation in these locations.

### 3.6 Integrated SMB

In this section, all major surface mass balance components are spatially integrated. We use both the ice masks native to each model as well as a common ice mask for this. Compared to RCMs, which are strongly forced by atmospheric reanalyses, our AMIP-style simulations experience relatively weak forcing at the ocean boundaries, which renders it unlikely that the actual

historical Greenland weather conditions are reasonably resolved. Furthermore, a 20-year model simulation is arguably not long enough to attain a robust mean climate. Hence, the numbers presented in Table 2 should be interpreted with some caution, as RACMO2 and CESM are not necessarily experiencing the same climate. The common ice mask is constructed based on the contiguous GrIS definition, as laid out by the PROMICE mapping project (Citterio and Ahlstrøm, 2013), which is bilinearly

upscaled from the 1 km RACMO domain to the respective CESM grids. In the remainder of this section, we will focus on the results that were obtained on the common ice mask.

GrIS-integrated precipitation is overestimated in all CESM simulations with respect to the RACMO2 regional model (Table 2). The bias in precipitation is largest for Uniform CESM (+186 Gt yr$^{-1}$, or +26 %) and reduces with increasing resolution to +89 Gt (+13%, VR-CESM55) and +38 Gt (+5%, VR-CESM28). This is in line with our earlier findings of progressive drying

with increased resolution discussed in Sections 3.2, 3.3, and 3.4. Melt, on the other hand, seems consistently underestimated in all CESM simulations (Table 2). The bias in total melt volume is smallest for coarse-resolution Uniform CESM (-148 Gt, or -29%) and largest for fine-resolution run VR-CESM28 (-205 Gt, or -40%). Melt is reduced by 47 Gt in VR-CESM55 and by 57 Gt in VR-CESM28, with respect to Uniform CESM. The majority of that is due to ice melt, which sees similar reductions of 42 Gt and 52 Gt, respectively (Table 2), with snow melt accounting for the remainder of 5 Gt in both cases. Refreezing volume

is comparable across the three different CESM simulations (Table 2), with only slightly higher numbers at higher resolution. These could be explained, for instance, by lower snow temperatures (greater "cold content") in these runs, which is consistent with the lower melt rates found. Surface runoff in CESM is the sum of bare ice melt and drainage from the bottom of the snow pack, i.e. liquid water originating from rain or melt that does not refreeze. Due to the reductions in total melt volume, runoff is also significantly reduced at higher resolutions (Table 2), leading to significant negative biases when compared against the

downscaled 1 km RACMO2 data. With respect to RACMO2, Uniform CESM underestimates runoff by 40 Gt (-13%), VR-CESM55 by 95 Gt (-32%), and VR-CESM28 by 114 Gt (-38%), which agrees with the reduction in ablation found in Section 3.5. Sublimation is enhanced in both VR runs compared to Uniform CESM (Table 2) which we attribute to higher 10 m wind speeds occurring in VR-CESM (not shown). GrIS sublimation in VR-CESM28 is 54 % higher than in Uniform CESM, and exceeds the RACMO2 figure by 8 Gt.

Overall, GrIS integrated SMB exceeds 500 Gt in all CESM simulations (Table 2), which is markedly more than the 376 $\pm$ 99 that RACMO2 estimates over the common mask. There appear to be two balancing factors. On one hand, precipitation is overestimated in all CESM runs, and more so in the runs at low resolution (Uniform CESM and VR-CESM55). On the other hand, runoff is underestimated in all CESM runs, and more so in the runs at high resolution (VR-CESM55 and VR-CESM28). However, the decrease in precipitation is larger than decrease in runoff, which means that the lowest integrated SMB value is

found in VR-CESM28 (521 $\pm$ 77 Gt yr$^{-1}$).

## 3.7   Drivers of runoff changes

In the previous sections, it was established that CESM reproduces the in-situ ablation zone measurements with less skill at higher spatial resolutions (Section 3.5) and that melt/runoff are increasingly underestimated (Section 3.6). Here, we further examine what is driving these regressions using both the grid cell mean output, as well as elevation class (EC) output that is

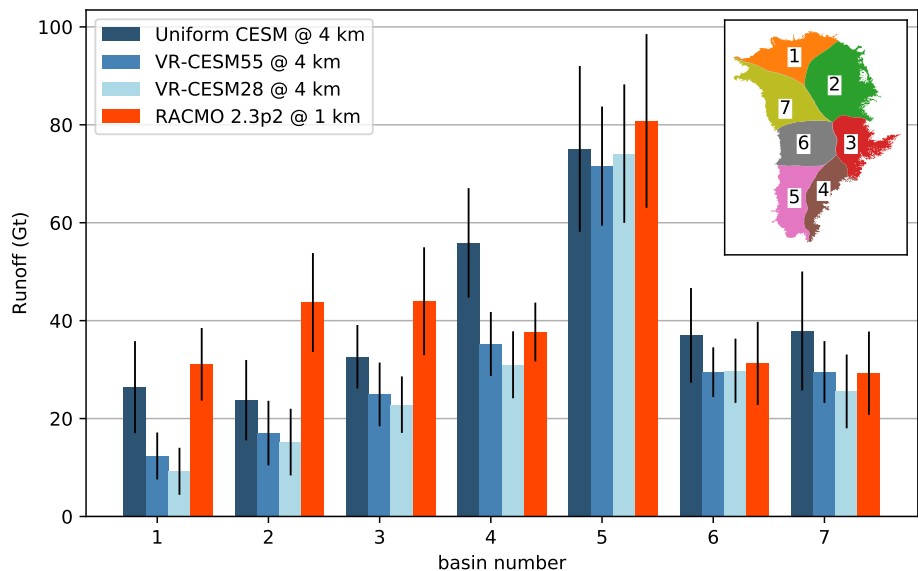

**Figure 8.** Mean basin-integrated runoff over the period 1980-1999. Error bars represent one standard deviation. CESM data have been manually downscaled down from their native resolution to 4 km using vertical SMB profiles generated by the elevation classes. For reference, RACMO2 downscaled runoff at 1 km resolution is shown. The extent of all basins combined equals the common ice mask in Table 2. Due to the manual interpolation, however, the total runoff for CESM does not match the value reported in Table 2.

manually downscaled to the CISM topography at 4 km using bilinear interpolation (for details on ECs, see Section 2.3). Note that this bilinear downscaling technique does not conserve mass and differs from the downscaling procedure inside the CESM coupler (Leguy et al., 2018).

First, we examine the spatial heterogeneity of runoff to uncover any regional differences. To this end, we aggregate the
downscaled runoff over 7 major GrIS drainage basins, derived from an ice flow mosaic updated from Rignot and Mouginot (2012), and use downscaled RACMO2 at 1 km as a reference. The results in Figure 8 indicate that Uniform CESM underestimates mean runoff in basin 1 (north), basin 2 (north-east), and basin 3 (east). In both VR runs, runoff decreases further in these regions and now falls outside of the standard deviation of RACMO2. In basin 4 (south-east), runoff is substantially overestimated in Uniform CESM (Figure 8), which can be explained through the poorly resolved precipitation field in Uniform
CESM. In reality, precipitation has steep gradients over this basin that are not resolved due to the coarse resolution (Figure 4). In both VR runs, precipitation shifts to lower elevations, which enhances meltwater buffering / refreezing and prevents bare ice exposure, two mechanisms through which runoff can be limited. Indeed, VR-CESM55 runoff is decreased and falls within one RACMO2 standard deviation in basin 4, whereas VR-CESM28 runoff seems slightly too low. The largest absolute runoff flux is found in basin 5 (south-west), which is equally well resolved by all CESM simulations, with integrated numbers that fall
within one standard deviation of the RACMO2 estimate. Finally, runoff in basin 6 (west) and basin 7 (north-west) is slightly overestimated in Uniform CESM, a bias that appears to be removed in both VR runs. In summary, this basin analysis indicates

that runoff is decreased across all GrIS basins, but with regional differences in magnitude. CESM underestimates runoff in the north (basin 1), north-east (basin 2) and east (basin 3) and this bias deepens with increasing resolution.

Next, we examine a number of atmospheric processes that could be driving the decreases in runoff. The results presented in Section 3.1 suggested that large-scale circulation changes are deemed to play a minor role. Still, temperature at the 700 hPa

pressure level (T700, linked to Greenland melt, e.g. Fettweis et al. (2013b)) is slightly lower in the VR-CESM simulations compared to Uniform CESM (Figure 9a). However, we note that the magnitude of this cooling does not match the much cooler surface temperatures (Figure 9b) and that the turbulent sensible heat flux is generally increased in VR-CESM, indicating that more heat is transferred to the surface in these simulations, not less (Figure S3). We hypothesize that the lower T700 in VR-CESM could be caused by the colder surface, rather than the other way around.

Instead, we argue that the observed decrease in runoff is driven by a combination of two main factors, and several feedbacks that relate to them. The first driver relates to a general decrease in GrIS cloud cover in VR-CESM, the associated cooling in the longwave radiative spectrum, and the notion that the thermal effect of clouds is crucial in triggering the onset of melt (Bennartz et al., 2013; Van Tricht et al., 2016b; Cullather and Nowicki, 2018). Figure 9c-e shows anomalies in VR-CESM surface elevation, cloud water path (CWP) and downwelling longwave radiation (LWd). The elevation anomalies appear similar in

both VR-CESM55 and VR-CESM28, with lower surface topography over the ocean and near the margins of the island, and with higher surface elevations inland (Figure 9c). Due to the higher and steeper terrain near the margins, orographic uplift and condensation are enhanced leading to increased cloud water path (CWP, the vertically integrated mass of liquid water and solid ice contained in clouds) with a decreased CWP higher up (Figure 9d). There are some exceptions, e.g. in north-east Greenland where locally CWP is reduced over the margin and ocean as well. Either changes in meso-scale flow driven by local

topography, or increased katabatic surface winds (Figure S2) are possible explanations for this. Due to the thermal effect of clouds, we find a strong correlation of LWd to CWP (Figure 9d-e). Both VR-CESM simulations show wide-spread decreases in LWd, including but not limited to the northern ablation zones, and we hypothesize that the onset of melt could be delayed in these sites. To some extent, LWd also correlates to skin temperature (Figure 9b), thereby providing a possible mechanism by which surface temperature are decreased in VR-CESM. We remark that the improved representation of topography, by itself,

does not lead to surface cooling, since ECs in CLM already account for differences between atmospheric topography and the actual ice sheet elevation (Section 2.3). Both a lower skin temperature – affecting snow ageing – and a delayed onset of melt are relevant controls on surface albedo and the associated albedo feedback. Figure 9f reveals positive JJA albedo anomalies up to 0.2 or more in VR-CESM, suggesting that this feedback is indeed active on Greenland, with the caveat that this plot may be severely impaired in places where ocean and land are mixed at 1 degree resolution (i.e. Uniform CESM), leading to

an artificial dipole pattern around the margins with negative anomalies over open ocean, and positive over land. The albedo feedback appears to be active over the adjacent sea ice as well (Figure 9f), since the fraction of sea ice is prescribed in these simulations, but we did not further investigate this.

The second driver relates to the EC subgrid downscaling (Section 2.3), which is argued to be less effective at compensating atmospheric biases at high spatial resolutions. We recall that the EC method in CESM has two mechanisms targeted to increase

melt in low-lying ablation zones, (1) a temperature lapse rate, which increases sensible heat transfer at low elevations, and

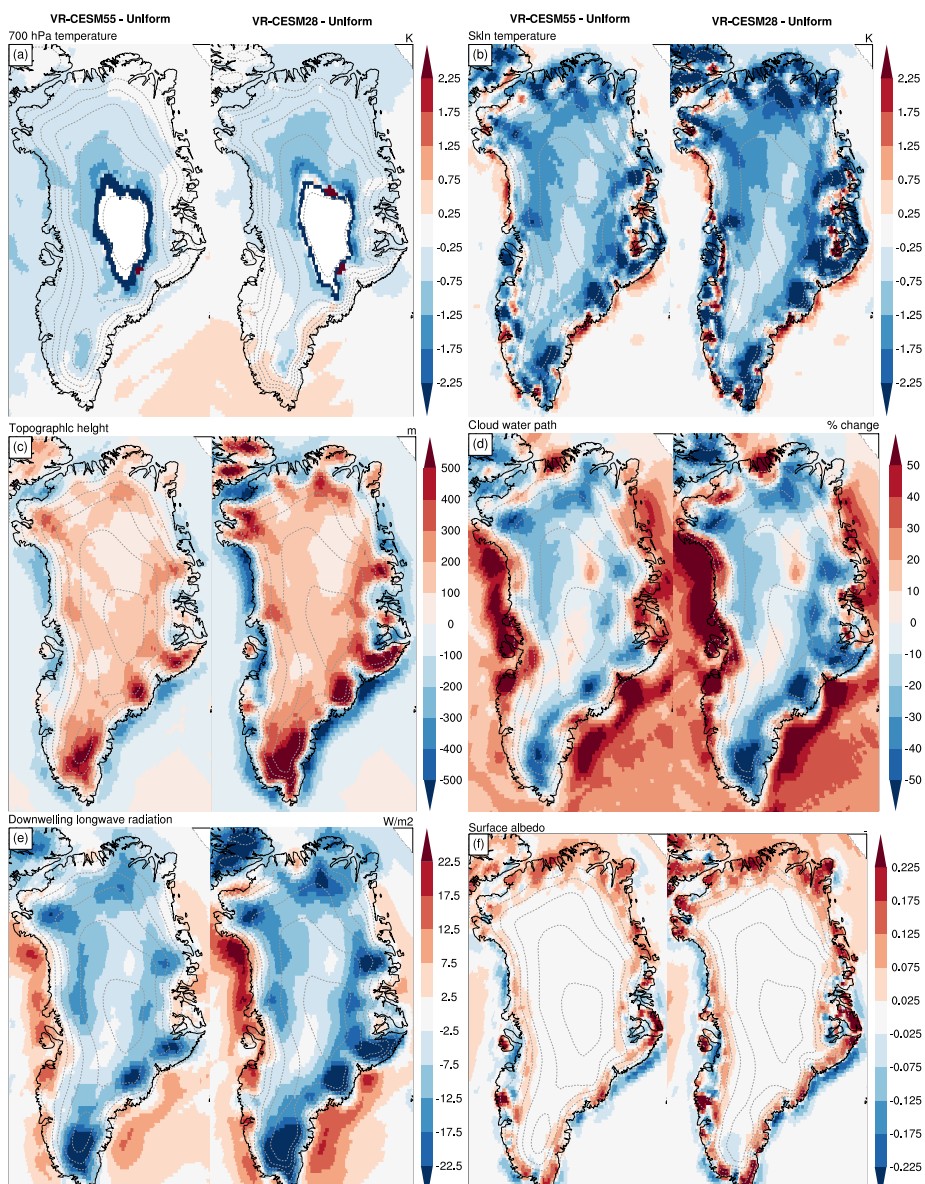

**Figure 9.** Summer (JJA) anomalies of atmospheric (CAM) variables over the period 1980-1999, relative to the coarse resolution reference simulation (Uniform CESM). Panel (a) 700 hPa air temperature [K], (b) radiative skin temperature [K], (c) CAM topographic height [m], (d) cloud water path [% change], (e) surface downwelling longwave radiation [W m$^{-2}$], (f) surface albedo [-]. Prior to subtraction, all data have been regridded to a common regular mesh of 0.25° using bi-linear interpolation. Therefore, these anomalies should be interpreted with some caution since they contain interpolations errors. The "sinking oceans" in panel (c) are explained by the smoothing operator applied to CAM topography, the imprint of which is much wider at low resolution than it is at high resolution (cf. Figure 2 and Section 2.2).

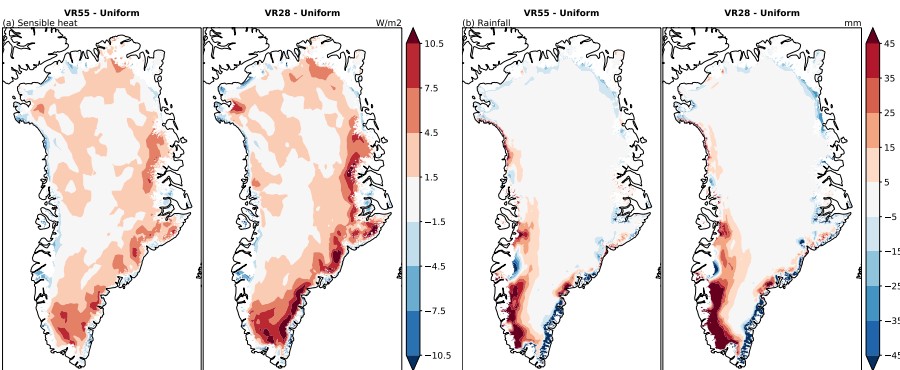

**Figure 10.** Summer (JJA) anomalies of (a) CLM sensible heat and (b) CLM rainfall over glaciated landunits, after downscaling, relative to the coarse resolution reference simulation (Uniform CESM) over the period 1980-1999. Data have been downscaled to 4 km using EC output.

(2) a downwelling longwave lapse rate, which does the same for LWd . Without these lapse rates, many GrIS ablation zones would not be resolved at all (Figure S1). The relevance of the EC downscaling depends, however, on the elevation variability actually present in a grid cell. Large variability (found at coarse grid spacing) means a sizeable difference between ablation zone and the grid cell mean elevation, leading to large corrections in sensible heat and LWd in low-lying ablation zones. Vice versa, a small variability in elevation (found at fine grid spacing) would lead to only minor corrections. Figure 10a depicts anomalies in downscaled sensible heat flux, and indeed we see a decrease in sensible heat over many low-lying ablation zones in VR-CESM, despite the higher grid cell means (Figure S3). Detecting a similar EC fingerprint is harder for LWd, due to the large cloud-induced signals in the anomaly maps (Figure S4). But besides these energy fluxes, another variable that is expected to be elevation dependent is rainfall, since the partitioning of snow and rain is recalculated in CLM based on temperature (Section 2.3) and temperature is EC-downscaled. Rainfall events add liquid water to the snow pack, thereby releasing latent heat, speeding up grain growth, and thus lower snow albedo (Oleson, 2013). We find that rainfall is reduced in VR-CESM across many ablation zones (Figure 10b), notably in the north and east, and is therefore likely to play a role in the reduction of melt and runoff in these locations.

### 3.8 Directions for further study

Our results underscore the notion that modelling GrIS ablation zones is a challenging task for a GCM, and that increasing the spatial resolution alone does not necessarily improve model skill. This ties in with Bacmeister et al. (2014), who remarked that increasing horizontal resolution by itself does not lead to dramatically improved climate simulations, and must be accompanied by new cloud and convection parametrizations. Existing parameterizations in CAM were developed with specific spatial and temporal scales in mind, and contain assumptions that may break down at higher resolutions (Bacmeister et al., 2014). Here, we shortly reflect on our findings from the previous sections and propose directions for future studies, with the aim of simulating a realistic Greenland surface climate at high spatial resolution.

Permanent snow cover over the northern tundras is a known model bias in this version of CESM and our results suggest that this bias worsens, rather than improves, on the VR refined grids (see e.g. 9f). This is an important model bias, with implications for surface temperature, albedo, and shortwave radiation over these areas (Figure S6-S8), possibly underpinning or reinforcing a general cold bias in northern Greenland. This bias may carry over to the GrIS, where cold tundra air might contribute to the weak sensible heat flux in north Greenland when compared to RACMO2 (Figure S9), although different surface wind speeds (Figure S10) play a role here as well. Further, it appears that the EC downscaling method is no longer effective in compensating for regional climate biases (Section 3.7) at higher resolutions, so future studies will need to address such climate biases directly.

On one hand, a further assessment of important atmospheric processes should be made, for instance the representation of super-cooled liquid clouds in CAM, and new parametrizations may be needed. Relevant metrics are cloud phase, frequency, and optical thickness. A recent study indicated that CAM5 simulates insufficient summer clouds when compared to observational data, in particular non-opaque liquid-containing clouds that have a strongly positive cloud radiative effect (Lacour et al., 2018). Next to their radiative properties, clouds have large control over the amount and phase of precipitation. We note that the three CAM-SE experiments presented in this study seem to underestimate the magnitude of rainfall over north Greenland, when compared to RACMO2 (Figure S11).

On the other hand, the representation of surface processes may need to be reviewed. There is reason to believe that the precipitation phase repartitioning currently implemented in CLM (Section 2.3) has a detrimental effect on the north Greenland simulation, where supercooled rain may be needed to darken snow and set off the melt-albedo feedback. Further, we note that CESM currently lacks drifting snow sublimation and erosion, which are important SMB factors in the relatively dry north of Greenland. To illustrate this, blowing snow was overestimated in a previous version of RACMO2 (RACMO 2.3p1) and caused too-wide ablation zones in the north (Noël et al., 2018). The three CESM experiments presented here all simulate north Greenland ablation zones which appears too narrow (Figure 7), which could be due to such missing processes. CLM snow physics could be another point of future development, as RCM studies have highlighted the importance of water percolation and sensitivity of melt to irreducible water content (e.g., van Angelen et al., 2012).

## 4 Summary and Conclusions

For the first time, regionally refined GCM simulations using VR-CESM have been performed at 55 and 28 km over the greater Greenland region to study the impact of spatial resolution on GrIS SMB. Compared to a uniform resolution (1° or ∼111 km) control run, topography is resolved with greater fidelity, leading to improved patterns in orographic precipitation, most notably in southern Greenland and along the western and eastern margins. At the same time, a general drying in the GrIS interior occurs, which substantially improves correlations to IceBridge accumulation radar and in-situ measurements of accumulation. Arguably, VR-CESM performs on-par with RCMs in reproducing these observations, especially at 28 km. GrIS integrated precipitation is reduced from 893 to 745 Gt in VR-CESM28, which is within 6% of a best-estimate RCM figure (707 Gt). The improved distribution of accumulation may prove pivotal in transient simulations, as snowfall modulates the timing and strength of the snow-albedo feedback (Picard et al., 2012) and impacts ice advection.

In the ablation zone, the CESM simulations were evaluated using geographically sparse in-situ measurements. Despite its coarse resolution of ∼111 km, we found that Uniform CESM reproduces these measurements to a reasonable degree, which represents a positive result for CESM at low resolution and suggests that the subgrid ECs are effective (Section 2.3, Figure S1). In both VR-CESM simulations, a positive SMB bias (i.e. too little ablation) developed in the ablation zone, which signals

a regression. This was reflected in GrIS-integrated runoff, which was found to be substantially lower in VR-CESM55 and VR-CESM28 compared to Uniform CESM and RACMO2. A basins-by-basin analysis revealed that the largest reductions in runoff are found in the northern and eastern basins, with a fairly good agreement in the other basins. The decrease in runoff is argued to be driven by two independent factors: (1) substantial reductions in LWd are found over large parts of the GrIS due to cloud redistribution, which is likely to delay the onset of melt; (2) a higher spatial resolution implies lower topographic

variability within a given grid cell, which renders the EC downscaling less effective in compensating for atmospheric biases in VR-CESM. Both these factors will induce the melt-albedo in a negative way, i.e. leading to higher albedo and further reduced melt, and are difficult to untangle from one another.

To conclude, our case study demonstrates that VR-CESM is a promising technique for dynamically downscaling GCM climate simulations over an Arctic region, while maintaining model consistency and allowing for feedbacks between the region

of interest and the rest of the globe. A finer resolution leads to better resolved storms that are taking different pathways than their low-resolution counterparts, and therefore change precipitation and cloud cover patterns on a local scale. VR-CESM can serve as a tool for modellers that are interested in the dynamical response of the GrIS to future SMB changes, at a reasonable computational cost. At the time this manuscript was written, it was not possible to run VR-CESM in coupled mode with an active ocean model. Still, high-resolution future projections of GrIS SMB could be generated using VR-CESM when high-

frequency output from a fully-coupled scenario simulation is used as a boundary conditions at the sea surface.

*Data availability.*  Climate model data used in our analysis are available on Zenodo, https://doi.org/10.5281/zenodo.2579606

*Author contributions.*  LvK and MRB originally conceived the study. The simulations were set up and carried out by LvK and AMR with help from ARH and CMZ. LvK led the analysis and the writing of the manuscript, with contributions from all the other authors.

*Competing interests.*  The authors declare no competing interests.

*Acknowledgements.*  The authors would like to thank the editor, Xavier Fettweis, and two anonymous referees for their constructive comments. Further, we had fruitful discussions with Paul Ullrich (UC Davis) and Aarnout van Delden (UU/IMAU), and the RACMO2 data was kindly provided by Brice Noël (UU/IMAU). This work was carried out under the program of the Netherlands Earth System Science Centre (NESSC), financially supported by the Ministry of Education, Culture and Science (OCW, Grantnr. 024.002.001). Author Alan Rhoades

was funded by the U.S. Department of Energy, Office of Science "An Integrated Evaluation of the Simulated Hydroclimate System of the Continental US" project (award no. DE-SC0016605).

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
