# Peer review of "Regional Grid Refinement in an Earth System Model: Impacts on the Simulated Greenland Surface Mass Balance"

_The Cryosphere, 2018_

## Referee Comment (RC1) · Anonymous Referee #1 · 17 Dec 2018

**Review of the Manuscript**
**"Regional Grid Refinement in an Earth System Model: Impacts on the Simulated Greenland Surface Mass Balance"**
**by Kampenhout et al.**

In their paper, Kampenhout and collogues investigate the impact of regional grid refinement on the Greenland surface mass balance. They compare three CESM simulations with regionally refined grids to a CESM simulation with a uniform grid. Further, they evaluate the model performance using remote sensing data, in-situ measurements as well as regional model simulations. They find, that a grid refinement improves accumulation patterns, however biases develop in the ablation zones. In order to explain differences in the ablation between the simulations they investigate differences in the atmospheric large-scale circulation and clouds. The manuscript is very interesting and well written and discusses a very relevant topic, which after major revision can provide a step forward in understanding resolution dependency of the surface mass balance.

**Major Comments:**
*Results and Limitations section (P.8-13 and 18):*
It should be appreciated that the error in the land cover input file is reported in the manuscript. Given the relatively short simulations in combination with the acceptable performance and throughput per day I was wondering why the authors did not consider to rerun the simulations. Although the error might not change the overall conclusions, a lot of absolute values are presented throughout the results section which are very likely to become different if the error was fixed.

*Results and Discussions:*
I am very skeptical towards not using the same ice sheet mask for a direct comparison between RACMO and the CESM simulations. The ice sheet extent has a significant influence on the Greenland integrated mass balance and specifically melt. Hence, I do not believe that a "fair" comparison can be made without using the same mask. As the authors focus on the low bias of melt in the VR-CESM simulations the authors should reconsider using an identical mask for the comparison; there is a good chance that the bias is a result of the smaller CESM ice mask. The mass conservation argument, as stated in their response to the editors comments, does not hold in my opinion. If the authors decide not to change the overall analysis in the result section, such an analysis should be included somewhere in the manuscript in order to estimate the effects of the mask on the integrated values.

**Minor Comments:**
*Abstract:* "The SMB in the accumulation zone is significantly improved compared …" – It should be added that the refinement leads to such improvement.

*Introduction/Model description:* It is not clear to me which version of CESM is used. This should be pointed out more clearly (CESM2 or CESM1) from the beginning.

*Page 4, Line 8:* Are these chosen constants a result of model tuning or used in other studies? Please add references. See also comment on the lapse rate below.

*Table 1:* The values for RACMO should be included for reference, as it would increase the readability and a direct comparison.

*Page 12, Line 29:* Please hypothesize which counteractive effects are acting.

*Page 14, Line 12:* ". ." – remove one .

*Page 14, Line 23:* Please state that a 0.25º mesh is chosen here.

*Page 17, Section "Latent Heat":* Regionally there are large differences and the response seems to be very dependent on the topography. The authors should be a little more detailed in their analysis.

*Page 17, Section "Subgrid downscaling":* Could this be tested by an additional simulation in which the lapse rate during the downscaling is slightly changed? Or can the authors think of another method to test this? See also the lapse rate comment above.

*Page 18, Line 23:* "&" to "and"

*Page 14, Line 32:* Have you investigated the blocking frequencies in the different simulations? Or is this a hypothesis?

---

## Referee Comment (RC2) · Anonymous Referee #2 · 18 Dec 2018

This paper describes a (currently unsupported) configuration of the CAM+CLM atmosphere/land models in a set of short AMIP-style experiments where regional grid refinements have been made over Greenland. The focus of the analysis is on the impact of the local grid refinements on the simulated surface mass balance of the Greenland ice sheet, by comparison with observations and a benchmark regional model. The paper is timely and generally clear and well written. Although useful already, I think it would benefit significantly from a few key changes, if the authors could manage them

Major comments:

Some of what I'd like to see improved is already acknowledged by the authors in section

5, Limitations. To start with, is the model so expensive that even one further run cannot be done to properly quantify the impact of the erroneous input file, even if you can't go further and rerun both the VR simulations with the correct boundary files? Section 2.1.5 suggests this should only take a couple of days. Another matter highlighted here that would seem to be possible to address is that all results of CESM vs RACMO in the Tables are presented as area-integrated, absolute quantities, although the area of the ice sheet used in each case is different. The difference in areas considered does make it difficult to know how far the anomalies wrt the RCM are really down to the CESM-VR physics, especially in the crucial, high ablation zones at the edges. I think some common area could be defined for analysis purposes, even without formally separating out the main sheet from the peripheral glaciers in CESM as section 5 states cannot yet be done.

More generally, from the analysis that has been done the paper doesn't come to any firm conclusion as to why there's apparently a systematic trend towards lower Greenland ablation with higher resolution, nor whether this really represents an improvement or degradation in model physics or overall performance. Going further, one would really like to see how the changes play out when the CISM ice sheet is coupled into the system. On pg18, there is a scope-limiting statement that has been left right to the end of the paper - if this paper is a limited, "exploratory first step" I think it that should be stated earlier to set up the readers expectations appropriately. I don't think these matters should really be out of scope for this study, but we should at least be warned earlier if the authors are going to declare that they are.

Minor comments

pg1,line7: "starts developing": clarify that the growing bias is a function of resolution rather than eg time.

pg1,line13: I don't think you need "relative"

pg3,line9: I wasn't sure until it was noted as a Limitation in section 5 that there really

had only been a single, twenty year run conducted for each CESM configuration shown here. Could that be made clearer, earlier?

pg5,line4: "similar to what is done in two-way coupled setups..." this isn't very helpful for readers unfamiliar with the CISM coupling. Could you rephrase, or add a reference?

pg14, line 12: two full-stops after "resolution"

section 4.1: how do the circulation anomalies shown compare with the variability in CESM? Could they be further compared with CESM-ERA biases for the relevant period, to judge whether the VR has improved matters? I think our Editor has previously suggested that 700mb is the best altitude to assess temperature biases for relevance to Greenland climate?

section 4.2: The above-suggested comparison with ERA could usefully be extended to cloud properties, and the surface fluxes could be compared with those from RACMO?

I don't think the content in sections 4.3 and 4.4 is really extensive enough to deserve their own separate headings?

pg19,line27: Even though the VR configuration isn't supported yet, is it worth adding a link to where people can get CESM, and/or lodging your configuration description with the paper?

---

## Author Comment (AC1) · 28 Feb 2019

**RC1**

In their paper, Kampenhout and collogues investigate the impact of regional grid refinement on the Greenland surface mass balance. They compare three CESM simulations with regionally refined grids to a CESM simulation with a uniform grid. Further, they evaluate the model performance using remote sensing data, in-situ measurements as well as regional model simulations. They find, that a grid refinement improves accumulation patterns, however biases develop in the ablation zones. In order to explain differences in the ablation between the simulations they investigate differences in the atmospheric large-scale circulation and clouds. The manuscript is very interesting and well written and discusses a very relevant topic, which after major revision can provide a step forward in understanding resolution dependency of the surface mass balance.

We thank the reviewer for his / her kind words and critical review. Below, we respond to his / her remarks in blue.

**Major comments**

*Results and Limitations section (P.8-13 and 18):*

It should be appreciated that the error in the land cover input file is reported in the manuscript. Given the relatively short simulations in combination with the acceptable performance and throughput per day I was wondering why the authors did not consider to rerun the simulations. Although the error might not change the overall conclusions, a lot of absolute values are presented throughout the results section which are very likely to become different if the error was fixed.

The reviewer raises a fair point. Since the other reviewer also raised this point we decided to re-run both VR simulations that were affected by the error. All tables, figures and numbers in the text have been updated accordingly.

*Results and Discussions:*

I am very skeptical towards not using the same ice sheet mask for a direct comparison between RACMO and the CESM simulations. The ice sheet extent has a significant influence on the Greenland integrated mass balance and specifically melt. Hence, I do not believe that a "fair" comparison can be made without using the same mask. As the authors focus on the low bias of melt in the VR-CESM simulations the authors should reconsider using an identical mask for the comparison; there is a good chance that the bias is a result of the smaller CESM ice mask. The mass conservation argument, as stated in their response to the editors comments, does not hold in my opinion. If the authors decide not to change the overall analysis in the result section, such an analysis should be included somewhere in the manuscript in order to estimate the effects of the mask on the integrated values.

We feel ambiguous on this point, and believe there is value in both. As a middle road, we have updated Table 2 with extra rows that display the integrated mass values over a common ice mask, but maintain the rows in which the components are integrated over the native mask. The analysis in the text uses primarily the common mask and has been updated accordingly.

**Minor comments**

*Abstract:* "The SMB in the accumulation zone is significantly improved compared ..." – It should be added that the refinement leads to such improvement.

*Thanks, we now start the sentence as follows: "On the refined grids, the SMB …"*

*Introduction/Model description:* It is not clear to me which version of CESM is used. This should be pointed out more clearly (CESM2 or CESM1) from the beginning.

*The version of CESM used is neither exactly CESM1 nor exactly CESM2. Indeed, it is an unsupported configuration of components, which gave the best results at the time. It uses the CAM5.4 atmosphere model (part of CESM1), with MG2 microphysics (part of CESM2). We believe that the main results should be reproducible using CESM2, as soon as CAM6 gives good results with variable resolution. At the start of our study, however, VR in CAM6 was still undergoing heavy testing.*

*For clarity, we've added this line to the introduction (P2 L33):*
*"The version of CESM used resembles the recently released CESM version 2 (CESM2), of which a more in-depth evaluation is planned in the near future."*

*Further, the following line was added to Methods (P3, L15):*
*"VR capabilities in CAM6, the new atmosphere model in CESM version 2 (CESM2, \url{www.cesm.ucar.edu/models/cesm2.0}) were still under beta testing at the start of our study, which explains the slightly older model version of CAM. "*

*Page 4, Line 8:* Are these chosen constants a result of model tuning or used in other studies? Please add references. See also comment on the lapse rate below.

*These are taken from the literature. Two references have been added.*

*Table 1:* The values for RACMO should be included for reference, as it would increase the readability and a direct comparison.

*We have done this.*

*Page 12, Line 29:* Please hypothesize which counteractive effects are acting.

*This sentence has been removed in the updated manuscript, since we did more analysis.*

*Page 14, Line 12:* ". ." – remove one .

*Done.*

*Page 14, Line 23:* Please state that a 0.25o mesh is chosen here.

*Done.*

*Page 17, Section "Latent Heat":* Regionally there are large differences and the response seems to be very dependent on the topography. The authors should be a little more detailed in their analysis.

*The analysis of latent heat was removed from the revised manuscript, since it is deemed not to be of first order importance.*

*Page 17, Section "Subgrid downscaling":* Could this be tested by an additional simulation in which the lapse rate during the downscaling is slightly changed? Or can the authors think of another method to test this? See also the lapse rate comment above.

*This section has been removed in the revised manuscript, since it was no longer deemed relevant. Just for completeness, there is currently a paper in preparation by Raymond Sellevold (TU Delft) specifically looking at the impact of different lapse rates on melt.*

*Page 18, Line 23:* "&" to "and"

*Done*

*Page 14, Line 32:* Have you investigated the blocking frequencies in the different simulations? Or is this a hypothesis?

*This was a hypothesis. Note that the Discussions section has been extensively rewritten in the revised manuscript.*

**RC2**

This paper describes a (currently unsupported) configuration of the CAM+CLM atmosphere/land models in a set of short AMIP-style experiments where regional grid refinements have been made over Greenland. The focus of the analysis is on the impact of the local grid refinements on the simulated surface mass balance of the Greenland ice sheet, by comparison with observations and a benchmark regional model. The paper is timely and generally clear and well written. Although useful already, I think it would benefit significantly from a few key changes, if the authors could manage them.

*We thank the reviewer for his / her kind words and critical review. Below, we respond to his / her remarks in blue.*

**Major comments**

Some of what I'd like to see improved is already acknowledged by the authors in section 5, Limitations. To start with, is the model so expensive that even one further run cannot be done to properly quantify the impact of the erroneous input file, even if you can't go further and rerun both the VR simulations with the correct boundary files? Section 2.1.5 suggests this should only take a couple of days.

*Following this comment and a similar comment by Reviewer 1, we decided to re-do both simulations that were affected by the erroneous input file. All tables, figures and numbers in the text have been updated accordingly.*

Another matter highlighted here that would seem to be possible to address is that all results of CESM vs RACMO in the Tables are presented as area-integrated, absolute quantities, although the area of the ice sheet used in each case is different. The difference in areas considered does make it difficult to know how far the anomalies wrt the RCM are really down to the CESM- VR physics, especially in the crucial, high ablation zones at the edges. I think some common area could be defined for analysis purposes, even without formally separating out the main sheet from the peripheral glaciers in CESM as section 5 states cannot yet be done.

*We feel ambiguous on this point, and believe there is value in both. As a middle road, we have updated Table 2 with extra rows that display the integrated mass values over a common ice mask, but maintain the rows in which the components are integrated over the native mask. The analysis in the text uses primarily the common mask and has been updated accordingly.*

More generally, from the analysis that has been done the paper doesn't come to any firm conclusion as to why there's apparently a systematic trend towards lower Greenland ablation with higher resolution, nor whether this really represents an improvement or degradation in model physics or overall performance. Going further, one would really like to see how the changes play out when the CISM ice sheet is coupled into the system. On pg18, there is a scope-limiting statement that has been left right to the end of the paper - if this paper is a limited, "exploratory first step" I think it that should be stated earlier to set up the readers expectations appropriately. I don't think these matters should really be out of scope for this study, but we should at least be warned earlier if the authors are going to declare that they are.

We agree with RC2 in that it was disappointing that the paper did not come to any firm conclusions on the trend towards lower ablation. We decided to extend our analysis accordingly and added Section 3.1 on large scale circulation and Section 3.8 on clouds and radiation. Still, there is some room for interpretation, which is left to Section 4. Discussions in the revised document.

Further, we feel that this paper should primarily remain concerned with SMB and the interpretation of atmospheric fields, and have not added any new analysis on the dynamical impact. We have added new scope limiting statements, notably:

P1. L9: "…pilot study.."
P2. L27: "… explore the impacts that the refinement has on GrIS SMB."
P21. L25: "… our case study…"

**Minor comments**

pg1,line7: "starts developing": clarify that the growing bias is a function of resolution rather than eg time.

Good point, we have added "with resolution".

pg1,line13: I don't think you need "relative"

OK, we removed this word.

pg3,line9: I wasn't sure until it was noted as a Limitation in section 5 that there really had only been a single, twenty year run conducted for each CESM configuration shown here. Could that be made clearer, earlier?

We added the following sentences to Sect. 2.1 (Modelling setup)
*"Three AMIP-style CESM simulations are performed over the years 1980-1999, a period prior to the onset of persistent circulation change and a strong decline in GrIS SMB in the 2000s \citep{Fettweis2013a, vandenBroeke2016}. Two CESM simulations are regionally refined, which makes that the resolution of the GrIS is different across all simulations (111, 55 and 28 km)."*

pg5,line4: "similar to what is done in two-way coupled setups..." this isn't very helpful for readers unfamiliar with the CISM coupling. Could you rephrase, or add a reference?

We've added the "CESM Land Ice Documentation and User Guide" as a reference, available online at https://escomp.github.io/cism-docs/cism-in-cesm/versions/release-cesm2.0/html/clm-cism-coupling.html

pg14, line 12: two full-stops after "resolution"

Thanks, this is fixed now.

section 4.1: how do the circulation anomalies shown compare with the variability in CESM? Could they be further compared with CESM-ERA biases for the relevant period, to judge whether the VR has improved matters? I think our Editor has previously suggested that 700mb is the best altitude to assess temperature biases for relevance to Greenland climate?

We have added Section 3.1 on large scale circulation, where we compare geopotential height and temperature at 500 hPa to ERA-Interim. Also, we have added Figure S4 in the Supplementary Material which does the same for JJA at 700 hPa.

section 4.2: The above-suggested comparison with ERA could usefully be extended to cloud properties, and the surface fluxes could be compared with those from RACMO?

In the newly added Section 3.8 we look at difference in cloud water path w.r.t. the reference simulation (Uniform CESM). We agree that clouds are an important subject for further study, and highlight clouds as a potential model weakness in Section 4. Discussions of the revised manuscript. Further, an evaluation of all surface energy fluxes using RACMO is a significant amount of work, and should involve the elevation class output from CESM. An evaluation of SEB is currently underway for the recently released CESM2, using CAM6 with the standard Finite Volume dynamical core.

I don't think the content in sections 4.3 and 4.4 is really extensive enough to deserve their own separate headings?

The number of separate headings has been reduced, including these.

pg19,line27: Even though the VR configuration isn't supported yet, is it worth adding a link to where people can get CESM, and/or lodging your configuration description with the paper?

We have added a link to the CESM2 web page in the Methods section. Further, our configuration is the result of many technical steps which were not documented in a manner appropriate for publication. Anyone that wishes to use VR-CESM is invited to contact the CESM Atmosphere Model Working Group liaison person for possibilities: http://www.cesm.ucar.edu/working_groups/Atmosphere/

[revised manuscript text omitted]

Contents of this file

[Figure]

Fig S1. Same as Figure 3, but for MAM

[Figure]

Fig S2. Same as Figure 3, but for SON

[Figure]

Fig S3. Same as Figure 3, but for DJF

[Figure]

Fig S4. Same as Figure 3, but for the 700 hPa pressure level in JJA. All grids cells that intersect with the 700 hPa level in any of the simulations have been excluded from the RMSE calculations.

[Figure]

Fig S5. Mean JJA terrestrial snow height (mmWE) in CESM over the period 1980-1999. The maximum snow height in CLM is 10,000 mmWE.

---

## Referee Report (RR1)

**Review of "Regional grid refinement in an earth system model: impacts on the simulated Greenland surface mass balance"**

by L. van Kampenhout et al.

**Summary:** The authors discuss the impact of spatial resolution on simulated Greenland ice sheet surface mass balance in a variable resolution version of the community earth system model (CESM). They find that enhancing the spatial resolution of the model over the Greenland ice sheet improved the spatial distribution of accumulation and reduced a positive bias relative to airborne remote sensing accumulation measurements at high elevations. The increased spatial resolution did not have much impact on patterns of atmospheric circulation, but led to decreased runoff along the east coast of Greenland. The authors attribute these changes to decrease in cloud water path, which led to changes in surface energy balance components, notably reduced longwave radiation. The authors also discuss potential reasons for an apparent positive bias in snow cover over tundra in northern Greenland in CESM which worsens with increasing spatial resolution.

**General Comments**

The paper is well-written, and the work is relevant, timely, and worthy of publication in *The Cryosphere*. I also feel that the authors have adequately addressed the concerns of the previous reviewers of the manuscript. I do have a few minor comments about the authors' interpretation regarding the role of clouds, and about the organization of sections at the end of the manuscript:

- 1. I feel the authors may be overemphasizing the role of clouds in the changes in runoff that occur along the Greenland east coast. The fairly large changes in longwave radiation that occur with increasing resolution extend far inland in southeast Greenland, while the net surface radiation anomalies are confined to the coast. Perhaps the longwave anomalies are balanced by similar downwelling shortwave anomalies. The spatial patterns of net surface radiation change actually seem to be better correlated with maps of surface albedo differences. I suggest the authors look at all components of the energy balance (e.g. for Figure 9) to see what the largest changes are and what might be contributing the most to the observed differences between simulations. (Less interesting figures could be included in the supplemental section.)
- 2. Surface albedo is discussed only briefly in the manuscript even though it is an important control on the surface energy balance. I suggest the authors consider the possibility of surface albedo changes while discussing some of the results.
- 3. The section labeled "discussion" seems a bit out of place, since it is mostly discussing the anomalies in snow cover over North Greenland tundra, and these differences are less relevant to the overall Greenland ice sheet mass balance, which is the focus of the rest of the paper. I suggest shortening this section, renaming it to reflect the snow accumulation issue, and including it as a subsection of Section 3. Perhaps Section 3 can also be renamed "results and discussion".

**Specific Comments**

- 1. P. 1, Line 5: Suggest mentioning precipitation or snowfall here to make clear what "wetting" and "drying" refer to.
- P. 2, Lines 6-16: The following recently published paper might also be interesting for the authors and could be mentioned here: Alexander, P. M., Legrande, A. N., Fischer, E., Tedesco, M., Fettweis, X., Kelley, M., Nowicki, S. M. J., and Schmidt, G. A. (2019) Simulated Greenland surface mass balance in the GISS ModelE2 GCM: Role of the ice sheet surface. *Journal of Geophysical Research: Earth Surface*, 124. https://doi.org/10.1029/2018JF004772.
- 3. P. 5, Line 12: Does the model include multiple layers? Briefly mention this here.
- 4. P. 5, Line 16 to P. 6, Lines 26-28: This part is a bit confusing. Initially it seems that the authors are saying that snow height was reset everywhere to 100 mm w.e. Perhaps revise to read something like: "Below 1774 m in elevation (which is roughly the present-day GrIS equilibrium line altitude), the initial snow amount was set to a minimum value of 100 mm w.e. to avoid snow cover hysteresis resulting from errors in the interpolated initial conditions." Also, if possible, please include a reference for the ELA being at 1774 m.
- 5. P. 9, Line 11: Where is the location of increased cyclonic flow?
- 6. P. 13, Line 3, Section 3.5: It could be mentioned here that elevation classes are used in all simulations, and therefore there is already "downscaling" occurring with respect to the surface model, which helps to explain why changing the spatial resolution does not substantially impact the comparison with ablation sites.
- 7. P. 13, Line 10: Suggest adding "for all model versions" after "right locations" for clarity.
- 8. P. 14 Lines 2-3: Could changes in surface albedo due also produce these changes in runoff? Does albedo vary across elevation classes? If not, a lower resolution could result in a lower coastal grid box elevation, increasing chances of bare ice exposure and lowering surface albedo (see Alexander et al., 2019). Perhaps the authors should mention this possibility, although the fact that changes are seen primarily along the east coast rather than the west coast suggests that albedo may not be so important.
- 9. P. 14, Line 17: This is a bit unclear. What is the melt underestimation a consequence of?
- 10. P. 15, Line 6: Is it being implied that lower snow temperatures are resulting in increased refreezing despite lower melt? Please clarify. Also suggest adding "higher" within the parentheses: '(higher "cold content")'.
- 11. P. 16, Line 13: What about surface conditions (e.g. surface albedo) or surface elevation differences for different resolution simulations? I suppose with the elevation class scheme active across all simulations, surface conditions may play a less important role in the differences, but the authors should mention this if this is the case.
- 12. P. 21, Lines 16-17: Indicate where in the ablation zone the biases developed; revise "too little ablation" to "too little ablation with increased resolution".

**Technical Corrections**

- 1. P. 4, Line 18: Add "were" before "scaled with horizontal resolution"
- 2. P. 6, Line 3: Change "off Figure 6" to "on Figure 6"
- 3. P. 6, Line 6: "SMB is the main focus..." This sentence seems out of place. Perhaps remove or move it?
- 4. P. 7, Line 33: Change "seasonal estimates" to "seasonal accumulation estimates" for clarity.
- 5. P. 8, Line 28: Should this read "one of the minima" instead of "one of the maxima"?
- 6. Figure 3: If possible include "anomaly" on the colorbar labels, e.g. "Z500 anomaly".
- 7. P. 8, Line 33-P. 9, Line 1: There does not appear to be any hatching in Fig. 3a-c. I think the authors can simply mention the hatching with regard to Figure 3f, and note that none of the other cases show any statistically significant difference. This can also be done for the caption of Figure 3.
- 8. P. 9, Lines 5-7: Define "JJA" and "MAM".
- 9. P. 9, Lines 9-10: There doesn't appear to be any hatching in Figures 3d and e. Revise to note only VR-CESM28 in Fig. 3f.
- 10. P. 10, Line 8: Remove "so" before "dominant".
- 11. P. 11, Line 7: suggest revising "the IceBridge radar data support" to "the comparison with IceBridge radar data supports".
- 12. P. 11, Line 17: Clarify that the r2 value is for the spatial correlation.
- 13. P. 11, Line 18: Revise to "regional model RACMO" or remove "regional"
- 14. P. 11, Lines 14-19: Change units to mm w.e. yr-1 for consistency here and throughout the manuscript.
- 15. P. 14, Line 5: Suggest adding "In this section..." before the start of the sentence for clarity.
- 16. Figure 6 caption: Mention that the shading shows the distribution of the differences.
- 17. Table 1: Again, change units to mm w.e. yr-1
- 18. P. 15, Line 19: Change "makes that" to "means that".
- 19. P. 16, Line 10: Add "in response to changes in resolution" after "summer" for clarity.
- 20. P. 17, Line 16: Suggest revising "negative longwave radiation" to "negative downwelling longwave radiation" for clarity.
- 21. P. 19, Line 23: I think "permanently increasing" should read "permanently decreasing".

---

## Author Response (AR2)

We thank the reviewers and editor for their thoughtful comments. Please find our responses in blue below.

**Editor report**

According to the suggestions from the 2 reviewers (who I thank for having greatly contributed to improve the manuscript), it is a pleasure for me to accept your paper for publication in TC. You have improved a lot your paper with this revised version.
Thank you, we feel the same way.

However, before final publication, reviewers suggested yet some minor changes and requested to improve your explanation of the "lower" runoff at the north of ice sheet at high resolution. I agree with them that your explanation about clouds is not convincing. According to your Fig S4, the warm biases of CESM at the north of ice sheet at 700hPa (it should be fine to check this also at 850hPa) is reduced when CESM is run at higher resolution. This is therefore just here an error compensation that explains why CESM_lowReso works better than CESM_highReso and not necessary linked to clouds. Your free atmosphere seems to be warmer in this area when CESM is run at lower resolution. To "prove" this, adding a figure showing Fig S4e - Fig S4d and S4f - Fig S4d could help. Comparing the JJA sensible heat flux should prove this?
We have attempted to address all reviewer concerns, see below. This led to some new analysis and updated plots. In particular, we now show T700 anomalies over Greenland in Figure 9a. However, we believe these (negative) anomalies are too small to explain the colder surface conditions (Figure 9b), like we explain in the text. Also, we find no reduction in sensible heat, but instead an increase (Figure S3). We look forward to hearing your opinion on the updated manuscript.

**Reviewer report #1**

This revision of the paper has addressed all of my previous concerns, I think it's a really useful piece of work now. A few things that occurred to me as I was reading through:

- the large-scale circulation comparisons in 3.1 are certainly necessary, but seeing as the results are pretty much null and what changes there are are statistically insignificant then illustrating them with four, multi-panel figures (1 in the main paper, 3 in Supplementary info) seems like overkill
Thanks you and good point, we have removed the three large-scale circulation figures from the Supplementary Material.

- if you are going to keep the Supplementary figures then please give them actual captions rather than making the reader refer back to the main paper for an explanation
The remaining Supplementary figures have been given full captions.

- the colour-scale used in figures 10 and S5 is a bit intense, especially since the majority of each figure is saturated at one end of the scale or another. Could these be recoloured so it's easier to make out the detail?

Figure 10 has been updated, and figure S5 has been removed from the updated manuscript.

- I suspect the diagnoses of the runoff biases in VR-CESM55 and VR-CESM28 presented in section 3.8 are correct, but it didn't seem obvious to me that the cloud and LW changes in figure 9b/c would impact the north and east (where the runoff bias is worst) more so than in the south. Figure 8 suggests that basin 4 on the south-east coast instead has a substantially reduced runoff bias. Any suggestions?

Yes, runoff is reduced in all regions. We decided to focus on basin 1-3 in the north and north-east, since here the differences to RACMO2 are largest and are becoming larger with increased resolution. The runoff comparison in basin 4 (south-east) is actually improved.

**Reviewer report #2**

**General Comments**

The paper is well-written, and the work is relevant, timely, and worthy of publication in *The Cryosphere*. I also feel that the authors have adequately addressed the concerns of the previous reviewers of the manuscript. I do have a few minor comments about the authors' interpretation regarding the role of clouds, and about the organization of sections at the end of the manuscript:

1. I feel the authors may be overemphasizing the role of clouds in the changes in runoff that occur along the Greenland east coast. The fairly large changes in longwave radiation that occur with increasing resolution extend far inland in southeast Greenland, while the net surface radiation anomalies are confined to the coast. Perhaps the longwave anomalies are balanced by similar downwelling shortwave anomalies. The spatial patterns of net surface radiation change actually seem to be better correlated with maps of surface albedo differences. I suggest the authors look at all components of the energy balance (e.g. for Figure 9) to see what the largest changes are and what might be contributing the most to the observed differences between simulations. (Less interesting figures could be included in the supplemental section.)

   Indeed, the longwave anomalies are balanced by similar downwelling shortwave anomalies, however they are modulated by albedo. It is generally hard to disentangle different drivers of melt, because all drivers impact the melt/albedo feedback. We have done some more analysis and included spatial maps of energy fluxes in the Supplementary Material. We also looked at energy fluxes downscaled to 4 km, something which we have not done before. This resulted in Figure 10 and an updated Section 3.7 which hopefully addresses your concerns.

2. Surface albedo is discussed only briefly in the manuscript even though it is an important control on the surface energy balance. I suggest the authors consider the possibility of surface albedo changes while discussing some of the results.
Yes, albedo is an important control on the SEB, but we believe albedo is "following", not driving, other changes in the SEB, and amplifies them. We tried to make this point clearer in the updated Section 3.7.

3. The section labeled "discussion" seems a bit out of place, since it is mostly discussing the anomalies in snow cover over North Greenland tundra, and these differences are less relevant to the overall Greenland ice sheet mass balance, which is the focus of the rest of the paper. I suggest shortening this section, renaming it to reflect the snow accumulation issue, and including it as a subsection of Section 3. Perhaps Section 3 can also be renamed "results and discussion".
Thank you, we agree that the discussion was not fully relevant to the main focus of the paper so we have done as you suggested. Section 3 is renamed "Results and Discussion" and we have added a new paragraph (3.8 Directions for further study) that contains a shortened version of the previous Discussion.

**Specific Comments**

1. P. 1, Line 5: Suggest mentioning precipitation or snowfall here to make clear what "wetting" and "drying" refer to. Done
2. P. 2, Lines 6-16: The following recently published paper might also be interesting for the authors and could be mentioned here:
Alexander, P. M., Legrande, A. N., Fischer, E., Tedesco, M., Fettweis, X., Kelley, M., Nowicki, S. M. J., and Schmidt, G. A. (2019) Simulated Greenland surface mass balance in the GISS ModelE2 GCM: Role of the ice sheet surface. *Journal of Geophysical Research: Earth Surface*, 124. https://doi.org/10.1029/2018JF004772.
Thanks, this reference has been added.
3. P. 5, Line 12: Does the model include multiple layers? Briefly mention this here.
Done, we have added ", with up to 12 layers."
4. P. 5, Line 16 to P. 6, Lines 26-28: This part is a bit confusing. Initially it seems that the authors are saying that snow height was reset everywhere to 100 mm w.e. Perhaps revise to read something like: "Below 1774 m in elevation (which is roughly the present-day GrIS equilibrium line altitude), the initial snow amount was set to a minimum value of 100 mm w.e. to avoid snow cover hysteresis resulting from errors in the interpolated initial conditions." Also, if possible, please include a reference for the ELA being at 1774 m. We changed the wording as follows: *"Below 1774 m in elevation (which corresponds to the highest GrIS elevation where SMB=0 in the RACMO2 climatology), the initial snow amount was set to a maximum value of 100 mm w.e. to avoid snow cover hysteresis resulting from errors in the interpolated initial conditions. A spinup simulation was then carried out to rebuild snow packs in CLM columns below this reset altitude, at least where CESM climate allows it."*
5. P. 9, Line 11: Where is the location of increased cyclonic flow? We added "over Greenland"

6.  P. 13, Line 3, Section 3.5: It could be mentioned here that elevation classes are used in all simulations, and therefore there is already "downscaling" occurring with respect to the surface model, which helps to explain why changing the spatial resolution does not substantially impact the comparison with ablation sites. We have changed the last sentence of this paragraph to read: *"Nonetheless, ablation zones are mostly predicted in the right locations (Figure 7), owing to the elevation class downscaling (Section 2.3) that is active in all simulations."*

7.  P. 13, Line 10: Suggest adding "for all model versions" after "right locations" for clarity. See point 6 above.

8.  P. 14 Lines 2-3: Could changes in surface albedo due also produce these changes in runoff? Does albedo vary across elevation classes? If not, a lower resolution could result in a lower coastal grid box elevation, increasing chances of bare ice exposure and lowering surface albedo (see Alexander et al., 2019). Perhaps the authors should mention this possibility, although the fact that changes are seen primarily along the east coast rather than the west coast suggests that albedo may not be so important. Albedo is allowed to vary independently in each elevation class (EC). Moreover, the height distribution of elevation across ECs also does not change, as the underlying hi-resolution topography remains constant. Changes in albedo are likely to occur, but must be externally forced and not due to the EC layout.

9.  P. 14, Line 17: This is a bit unclear. What is the melt underestimation a consequence of? Sorry, we meant to say "consistently". The sentence is updated to: *"Melt, on the other hand, is consistently underestimated in all CESM simulations"*

10. P. 15, Line 6: Is it being implied that lower snow temperatures are resulting in increased refreezing despite lower melt? Please clarify. Also suggest adding "higher" within the parentheses: '(higher "cold content")'. Yes, this is what we mean. We note that refreezing of rain is also a non-negligible term in CESM. We added the word "higher" for clarity, as suggested.

11. P. 16, Line 13: What about surface conditions (e.g. surface albedo) or surface elevation differences for different resolution simulations? I suppose with the elevation class scheme active across all simulations, surface conditions may play a less important role in the differences, but the authors should mention this if this is the case. We have elaborated our analysis on elevation classes and albedo in the updated Section 3.7.

12. P. 21, Lines 16-17: Indicate where in the ablation zone the biases developed; revise "too little ablation" to "too little ablation with increased resolution". We have updated this sentence to: *"In both VR-CESM simulations, a positive SMB bias (i.e. too little ablation) developed (…) "* which is hopefully more clear.

**Technical Corrections**

1. P. 4, Line 18: Add "were" before "scaled with horizontal resolution" Done
2. P. 6, Line 3: Change "off Figure 6" to "on Figure 6" Done

3. P. 6, Line 6: "SMB is the main focus..." This sentence seems out of place. Perhaps remove or move it? The sentence has been removed

4. P. 7, Line 33: Change "seasonal estimates" to "seasonal accumulation estimates" for clarity. Done

5. P. 8, Line 28: Should this read "one of the minima" instead of "one of the maxima"? Yes, fixed

6. Figure 3: If possible include "anomaly" on the colorbar labels, e.g. "Z500 anomaly". Done

7. P. 8, Line 33-P. 9, Line 1: There does not appear to be any hatching in Fig. 3a-c. I think the authors can simply mention the hatching with regard to Figure 3f, and note that none of the other cases show any statistically significant difference. This can also be done for the caption of Figure 3.

We updated the caption as follows: "*Hatching in panel f indicates that the VR-CESM simulation is significantly different (p < 0.05) from Uniform CESM. No significance was found in panels b, c, and e.*"

8. P. 9, Lines 5-7: Define "JJA" and "MAM". Done

9. P. 9, Lines 9-10: There doesn't appear to be any hatching in Figures 3d and e. Revise to note only VR-CESM28 in Fig. 3f. We have removed the statement on hatching from the main text.

10. P. 10, Line 8: Remove "so" before "dominant". Done

11. P. 11, Line 7: suggest revising "the IceBridge radar data support" to "the comparison with IceBridge radar data supports". Done

12. P. 11, Line 17: Clarify that the $r^2$ value is for the spatial correlation. We rewrote the sentence to: "*At the same time, the spatial correlation is substantially enhanced (r^2, Table 1*"

13. P. 11, Line 18: Revise to "regional model RACMO" or remove "regional" Done, we removed the word 'regional'

14. P. 11, Lines 14-19: Change units to mm w.e. $yr^{-1}$ for consistency here and throughout the manuscript. Done

15. P. 14, Line 5: Suggest adding "In this section..." before the start of the sentence for clarity. Done

16. Figure 6 caption: Mention that the shading shows the distribution of the differences.

The caption has been updated to: "*Point-by-point SMB differences between model and reference observations. Shading indicates the distribution, and horizontal line segments indicate maximum, median, and minimum value.*"

17. Table 1: Again, change units to mm w.e. $yr^{-1}$ Done

18. P. 15, Line 19: Change "makes that" to "means that". Done

19. P. 16, Line 10: Add "in response to changes in resolution" after "summer" for clarity. Done

20. P. 17, Line 16: Suggest revising "negative longwave radiation" to "negative downwelling longwave radiation" for clarity. Done

21. P. 19, Line 23: I think "permanently increasing" should read "permanently decreasing". Yes, thank you, this has been fixed

[revised manuscript text omitted]

---

## Author Response (AR3)

**Editor Comments**

Thanks for this revised version and for your clear and honest explanation of the performance "decrease" of CESM when it is run at higher resolution.

For me, the downscalling technique has originally been developed (and probably tuned) to compensate in part biases when CESM is run at low resolution. As you explain now well, when CESM is run at high resolution, the corrections brought by this downscalling technique are not enough to compensate the CEMS biases. This downscalling technique is particular dependent of the (fixed) vertical lapse rates used to extrapolate TT, LWD, ... to the sub-grid topography. Using other (in fact larger) values for these lapse rates when CEMS is run at higher resolution could fix in part the "decrease" of performance. If you agree with me, feel free to mention this in your conclusion.

Anyway, your paper is ready for me in the present state and could be sent to the Copernicus office for the typesetting.

Thank you, we are glad that you liked it. We feel that although the EC downscaling technique is adequate at coarse resolution, it should no longer be needed at higher resolutions. Indeed, larger lapse rates could increase runoff, but this may be for the wrong reasons and potentially lead to strange dSMB/dZ.  We decided not to make any changes to the paper anymore.